# Degron masking outlines degronons, co-degrading functional modules in the proteome

Mainak Guharoy [1,2,3 ✉], Tamas Lazar [1,2], Mauricio Macossay-Castillo[1,2] & Peter Tompa [1,2,4 ✉]

Effective organization of proteins into functional modules (networks, pathways) requires systems-level coordination between transcription, translation and degradation. Whereas the cooperation between transcription and translation was extensively studied, the cooperative degradation regulation of protein complexes and pathways has not been systematically assessed. Here we comprehensively analyzed degron masking, a major mechanism by which cellular systems coordinate degron recognition and protein degradation. For over 200 substrates with characterized degrons (E3 ligase targeting motifs, ubiquitination sites and disordered proteasomal entry sequences), we demonstrate that degrons extensively overlap with protein-protein interaction sites. Analysis of binding site information and protein abundance comparisons show that regulatory partners effectively outcompete E3 ligases, masking degrons from the ubiquitination machinery. Protein abundance variations between normal and cancer cells highlight the dynamics of degron masking components. Finally, integrative analysis of gene co-expression, half-life correlations and functional relationships between interacting proteins point towards higher-order, co-regulated degradation modules ('degronons') in the proteome.

[1] VIB-VUB Center for Structural Biology, Pleinlaan 2, 1050 Brussels, Belgium. [2] Structural Biology Brussels, Department of Bioengineering Sciences, Vrije Universiteit Brussel, Pleinlaan 2, 1050 Brussels, Belgium. [3] VIB Bioinformatics Core, Technologiepark-Zwijnaarde 75, 9052 Ghent, Belgium. [4] Institute of Enzymology, Research Centre for Natural Sciences of the Hungarian Academy of Sciences, 1117 Budapest, Hungary. ✉email: mainak.guharoy@vib.be; peter.tompa@vub.be

Precise control over the abundance of proteins is critical for cellular homoeostasis and the regulation of cellular pathways. In vivo protein abundance, often thought to be primarily regulated by gene transcription and mRNA translation[1], is also markedly affected by protein degradation[2,3], specifically regulating biological pathways[4,5]. Synergy between transcription, translation and degradation is important for the operation of proteome modules (circuits, subnetworks, pathways), as demonstrated by transcriptional networks organized into modules of co-expressed genes[2,6]. Previous proteome-wide studies showed that functionally related proteins have correlated turnover and similar half-lives[1,7–9], possibly due to mutual stabilization of the interaction partners and/or the masking of their proteolytic determinants[10,11]. Clearly, more work is required to characterize the systems-level coordination of protein degradation.

The ubiquitin-proteasome system (UPS) regulates biological processes via targeted degradation[12]. Target protein (substrate) selection is based on the specific recognition of degradation elements (degrons) by E3 ubiquitin ligases[13]. Substrate degrons display a tripartite hierarchy[14] consisting of a primary degron (a sequence motif that recruits an E3 ligase), secondary degron (sites of substrate ubiquitination, Ubsites), and tertiary degron (an intrinsically disordered region, IDR, that facilitates engagement and initial substrate unfolding by the proteasome) (Fig. 1a, b). Cooperative and successive action of all three degron elements enables the specific targeting and degradation of most UPS substrates. Previously, we observed that degrons are preferentially located within IDRs of substrates[14,15]. IDRs potentially enable interactions with multiple partners[16], being enriched in interaction motifs and posttranslational modification (PTM) sites that modulate binding preferences[17]. We reasoned that if binding sites for alternative partners overlap with degrons, the resulting competing interactions would mask and interfere with degron function, preventing their recognition by UPS components (Fig. 1c). Thereby, the structural plasticity of IDRs that enable embedding a dense network of overlapping interaction sites, can regulate protein turnover and exert dynamic control over the coordination of cellular pathways.

In fact, several studies have demonstrated that protein-protein interactions (PPIs) stabilize interacting proteins and coordinate their turnover. Degron masking was observed among transcription factors (TFs), for example, the yeast transcriptional repressors, MATa1 and MATα2[18]. In fact, MATα2 possesses multiple degrons that all overlap with corepressor binding sites[19]. E2F1, an important cell cycle regulatory TF, is stabilized by degron masking upon binding to the retinoblastoma tumor suppressor protein[20]. Among C/EBP TFs, NF-IL6 (C/EBPβ) is stabilized by homodimerization, and C/EBPγ and C/EBPζ are stabilized upon heterodimerization with NF-IL6[21]. Other important examples include β-catenin and p53, both of which are stabilized against proteasomal degradation by multiple PPIs, with important functional consequences (Supplementary Table 1). The overlap of

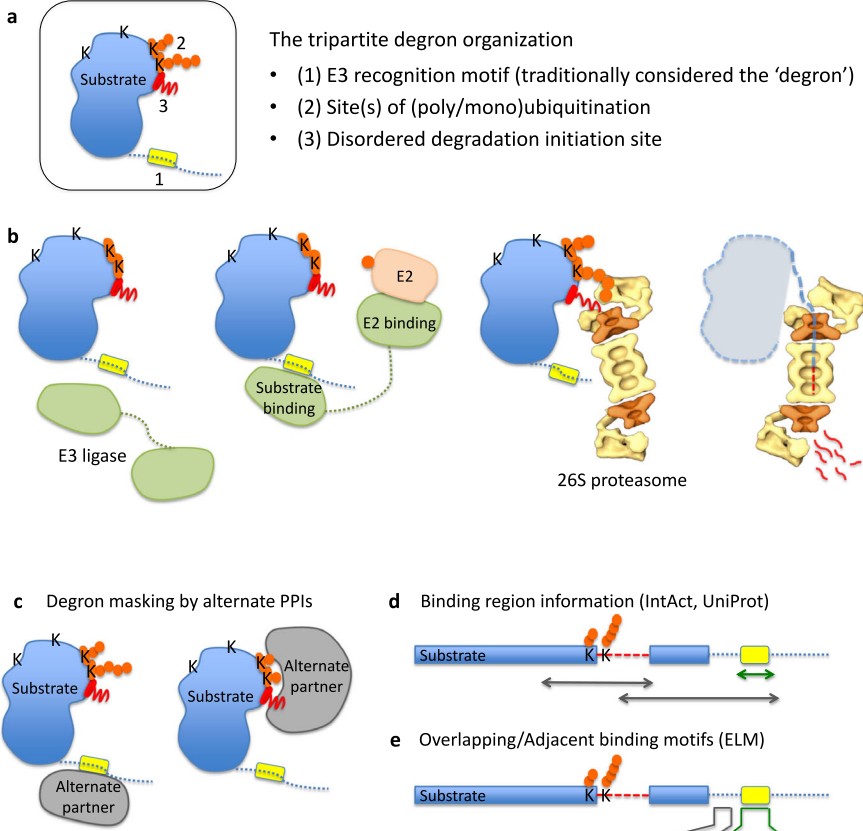

**Fig. 1 Schematic of tripartite degrons and their masking by PPIs. a** Tripartite degron organization, and, (**b**) schematic overview of how each of the degron components mediates specific steps in substrate selection and degradation in the UPS (adapted from Guharoy et al.[14]). Firstly, E3 ligases target specific substrates via primary degrons, followed by ubiquitination of single (or multiple, neighboring) lysines, K, by complexes consisting of E3 ligase and appropriate E2 conjugating enzyme(s). Ub-tagged substrates are then targeted to the 26 S proteasome for degradation. Tertiary degrons (located within or near Ubsites) are IDRs that initiate substrate unfolding and entry into the proteasomal core. **c** Alternate partners, APs, can bind to substrate segments harboring degron element(s), masking them from the UPS machinery. Binding site data analyzed in this study were obtained from (**d**) IntAct[24], UniProtKB[33], and (**e**) motif data from the Eukaryotic Linear Motif (ELM) resource[35].

degrons with regions of the protein involved in multimer assembly, such as transmembrane domains, is also a common feature in the quality control of secretory and membrane proteins: unassembled subunits undergo ER-associated degradation (ERAD)[22,23].

Although these examples hint at widespread tuning of degron accessibility by PPIs, degron masking has not been systematically assessed. Here we used our previously assembled datasets of experimentally validated degrons[14], onto which we mapped experimentally derived partner binding sites to explore potential degron masking (Fig. 1d, e). This study allowed us to understand how PPIs can interfere with degron function and influence protein stability. We also analyzed protein abundances to assess binding competition between E3s and degron masking partners and compared degron masking complexes in normal and disease cells. Finally, to achieve a proteome-scale perspective, we analyzed the *Saccharomyces cerevisiae* interactome by integrating multiple datasets (protein half-lives, abundances, gene co-expression and gene ontology data). Physically interacting proteins had correlated half-lives, mRNA co-expression and showed functional similarity. We propose that degron shielding by PPIs results in mutually correlated turnover (degradation) profiles between functionally connected, physically interacting groups of proteins (we termed such functionally interconnected PPI modules as "degronons"), highlighting the importance of this basic regulatory mechanism in cellular proteostasis.

## Results

**Protein-protein interactions masking degron elements**. To assess the effect of PPIs on protein stabilities and to dissect mechanisms, firstly we performed an extensive literature review and identified examples (Supplementary Table 1) where PPIs were demonstrated to inhibit degradation and increase protein half-lives. The proteins were diverse, both structurally and functionally, highlighting the potentially broad scope of this regulatory phenomenon. However, in most cases, the nature and location of degron(s) and the regions where stabilizing partners bind, remained unidentified. Therefore, we asked if stabilization resulted from partner proteins binding to sites overlapping substrate degrons, thus masking them from the UPS machinery (Fig. 1c). To answer this question systematically, we used our previously collected datasets of UPS substrates with experimentally annotated degrons[14], onto which we projected experimentally derived PPI binding site information (Fig. 1d), obtained from the IntAct database[24]. As illustrated in Fig. 1a, b, degrons are constituted of primary, secondary and tertiary elements[14]. Primary and secondary degrons were experimentally determined, whereas tertiary degrons were predicted as the IDR nearest in sequence to each Ubsite and containing at least 20 consecutive disordered residues[14]. We based this definition on studies into the initiation of degradation of ubiquitinated substrates by the 26 S proteasome[25].

Many substrates had PPI partners whose annotated binding sites overlapped with substrate degrons (Fig. 2a–c). Our dataset contained 157, 42 and 34 proteins with annotated primary, secondary and tertiary degrons, respectively (Supplementary Data 1); information about binding partners were available for 136, 40 and 33 proteins respectively, and binding site annotations were found for 89, 32 and 27 proteins. UPS-related partners were filtered out based on GO annotations (Supplementary Data 2), to identify bona fide stability-modulating partners that participate in functions other than degradation targeting. After filtering, degron-overlapping binding sites were identified for 62, 26 and 24 substrates, corresponding to the three degron categories. For certain substrates, highlighted on Fig. 2a–c, the proportion of

these degron masking, alternative partners (APs) amount to a sizeable fraction of the total number of partners, indicating tight and prevalent degron regulation by PPI-based masking. These substrates included highly connected (hub) proteins (e.g., human p53 (gene name: TP53), androgen receptor, p27, p21, c-Jun, β-catenin, HIF1α; Fig. 2a–c).

Figure 2d–f show specific examples where structural data enables us to visualize degron masking. The disordered N-terminal transactivation domain (TAD) of p53 contains the primary degron (19FSDLWKLL[26]) targeted by the E3 Mdm2/Hdm2. This degron is masked in the interface between p53 TAD and the nuclear receptor coactivator binding (NCBD) domain of CREB-binding protein, CBP[26] (Fig. 2d). Moreover, p53 TAD binds several additional domains of CBP (TAZ1, TAZ2 and KIX) and the CBP paralog, p300, as well as other APs. TAD:TAZ1 and TAD:TAZ2 complexes similarly demonstrated the bipartite TAD interface, masking the Mdm2-binding degron[27] (Supplementary Fig. 1). Upon DNA damage, multisite phosphorylation of p53 on Ser15, Thr18 and Ser20 significantly reduce binding affinity to Hdm2, and multiple CBP/p300 domains successfully compete for the p53 TAD[28]; thus, degron masking stabilizes p53 and contributes to transcriptional stress response by enhanced CBP/p300 recruitment.

Figure 2e illustrates masking of a secondary degron: the degradation-linked Ubsite (Lys97) of the transcriptional coactivator YAP1 is masked in the complex with TEAD1. Of three distinct YAP1 segments that wrap around TEAD1, the segment 86–100 (containing Lys97) is most crucial for complex formation[29]. Although its side chain points outwards from the surface, Lys97 would have significantly lower backbone flexibility in the complex, reducing productive nucleophilic attack on E2~Ub conjugates. Moreover, steric hindrance should preclude access by E3/E2 complexes, preventing ubiquitination. The tertiary degron of YAP1 is also the region 80–171 (Supplementary Fig. 2a, b), partly masked by TEAD1. The homologous TEAD2 and TEAD4 also mask these degrons (Supplementary Fig. 2c). TEAD TFs have been implicated in oncogenic functions of YAP[30] and stability regulation by degron masking may be a conceivable mechanism.

Concerning the tertiary degron, the majority of experiments had been performed using model substrates, engineering combinations of structured domains and IDRs[25,31]. Tertiary degrons in our dataset were all predicted as the IDR nearest in sequence to each experimental Ubsite. Many of these IDRs contain annotated partner-binding sites (Fig. 2c), suggesting that certain interactions might inhibit degradation by reducing accessibility of the tertiary degron for proteasomal engagement. In a recent study, the N-terminal IDR of yeast Mdy2 was demonstrated to perform the role of the tertiary degron in a physiological setting[32]. However, in complex with Get4, this IDR is masked (Fig. 2f) and Mdy2 is significantly stabilized[32].

To better understand the functional relevance of degron masking, the masking partners were analyzed to see whether (and to what extent) they exhibit any tendency towards functions similar to that of the substrates whose degradation they potentially regulate. Taking the proteins analyzed in Fig. 2, we quantified (using Gene Ontology Biological Process term comparisons, as described in Methods) the pairwise functional similarity between substrate-partner pairs and the subset of substrate-degron masking partner pairs (Supplementary Fig. 3). We expected both groups to exhibit functional similarity, since these are all high-confidence, experimentally verified, physically interacting proteins. Hence, they must share certain similar cellular functions and participate in common pathways. Interestingly, the degron masking partners showed significantly higher functional similarity to their corresponding substrates (whose degrons they mask) as compared to general interaction partners

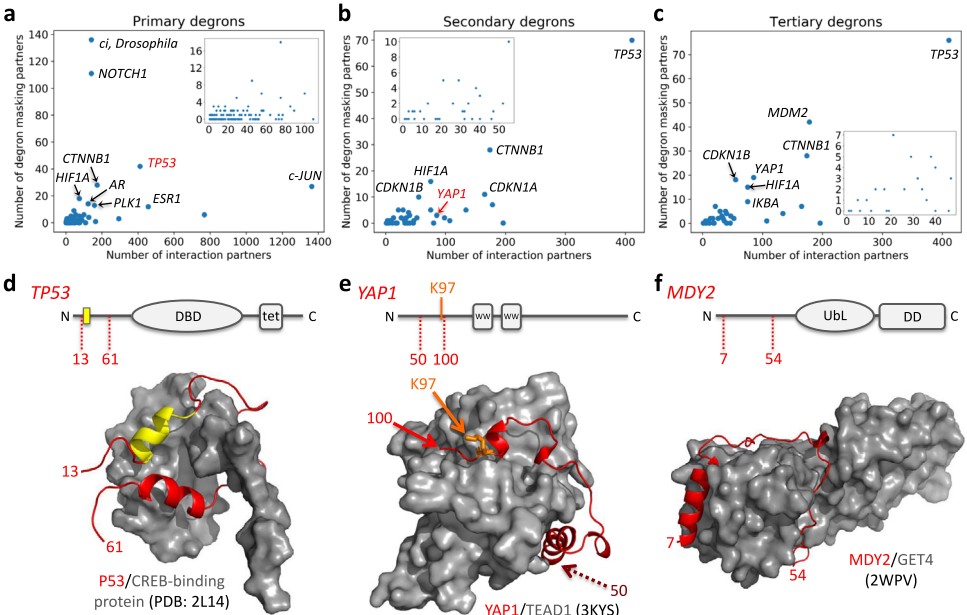

**Fig. 2 Degron masking by PPIs.** Plots showing the total number of experimentally identified PPI partners from IntAct (each data point corresponds to one substrate) versus the number of partners whose known binding site overlaps with (**a**) primary, (**b**) secondary, and (**c**) tertiary degrons. Substrates for which at least 10 degron masking APs were identified are labeled. The insets show a zoom-in view of the clustered data points at the bottom left of each plot. **d**–**f** Examples from each degron category where available structural data showed the degron containing segment in complex with a non-UPS masking partner, AP. Domain diagrams showing the location of the degron element and the substrate segment (in red) present in the crystal structure of the substrate-AP complex; drawn with PyMol (https://pymol.org/).

of those substrates (Supplementary Fig. 3). Therefore, this strongly indicates that protein stability is extremely well regulated and that degron masking partners must constitute, in a sense, a 'special' subset of a substrate protein's interactome. Furthermore, this observation was consistent for the masking partners of all three degron types, strongly suggestive of the relevance of masking all degron elements (primary, secondary, and tertiary).

Of note, certain PPI partner(s) may mask multiple degrons simultaneously, especially when they are located proximally, as illustrated by β-catenin and HIF1α (Supplementary Figs. 4, 5). IntAct binding site annotations are often obtained from deletion experiments used to map binding domains or from mutational studies highlighting the contributions of specific residues (Supplementary Fig. 6a). The length distribution of degron-overlapping binding segments showed that segments <100 AA were most prevalent, although longer segments were also found (Supplementary Fig. 6b). Longer segments may reflect limitations in binding-site resolution, but also raise the possibility of masking multiple degrons, as alluded to above. In addition to IntAct[24], we queried UniProtKB[33] for curated annotations such as binding sites, functional sites, PTMs and other relevant biochemical information indicating physical degron masking and functional interference (Supplementary Data 3–5). Previously, we showed that degrons are preferentially present in substrate IDRs[14]. These observations emphasize that by utilizing IDRs to embed degrons, unique IDR properties, such as structural adaptability that enables multi-partner binding can result in molecular mechanisms for regulating protein stability.

**Short interaction motifs overlapping with degrons enable regulation of protein stability.** Next, we analyzed the overlap of degrons with annotated functional protein motifs, since motif data provide a precise delineation of PPI sites (Fig. 1e). Since primary degrons constitute a prominent class of Short Linear Motifs (SLiMs)[34] (short, often intrinsically disordered functional sites typically 3–20 residues in length; also called Eukaryotic

Linear Motifs, ELMs), we queried the ELM database[35] for motifs that overlap with degrons (see Methods). We demonstrate here how multiple, clustered motifs act as input sites for signal integration and enable complex regulatory decisions on substrate degradation. Depending on their functional type and context, overlapping motifs influence degron function by providing binding sites for negative regulators (the interaction initiates/ enables substrate degradation) or positive regulators (whose binding prevents degradation) of half-life. Although experimentally verified ELMs remain limited[36] which precludes a statistical analysis, nevertheless, we found numerous degron overlapping ELMs (Fig. 3a; Supplementary Data 6–8). Qualitatively, therefore, it is clear that all three degron elements are potentially impacted by these overlapping ELMs and the proteins that they recruit, and as more ELMs are identified over time, further cases will be identified.

Modification and docking motifs were the most common (Fig. 3a): docking motifs recruit enzymes (e.g., kinases) that modify the substrate at nearby modification motifs. The latter, depending on their single or combinatorial PTM status, create ultrasensitive interaction switches[17,34]. Kinases including cyclin-dependent kinases (CDKs), PLK-1 and GSK3β can negatively regulate half-life, phosphorylating and activating substrate phosphodegrons, which are then recognized by SCF E3s[37]. Conversely, kinases also function as positive regulators in cases where phosphorylation, either of the degron or its neighborhood, prevents degron recognition by forming phospho-inhibited modules[38]. An example is Cdc6, an essential licensing factor involved in DNA replication origin firing. Adjacent to its KEN degron are a cyclin docking motif and a Cdk2 phosphorylation site (Fig. 3b). CDKs promote licensing by employing these motifs to phosphorylate Cdc6 upon cell cycle reentry following quiescence; phosphorylation stabilizes Cdc6 by preventing its association with APC/C(Cdh1)[39].

Figure 3a highlights the importance of docking and modification sites not only for primary degrons, but also for regulating

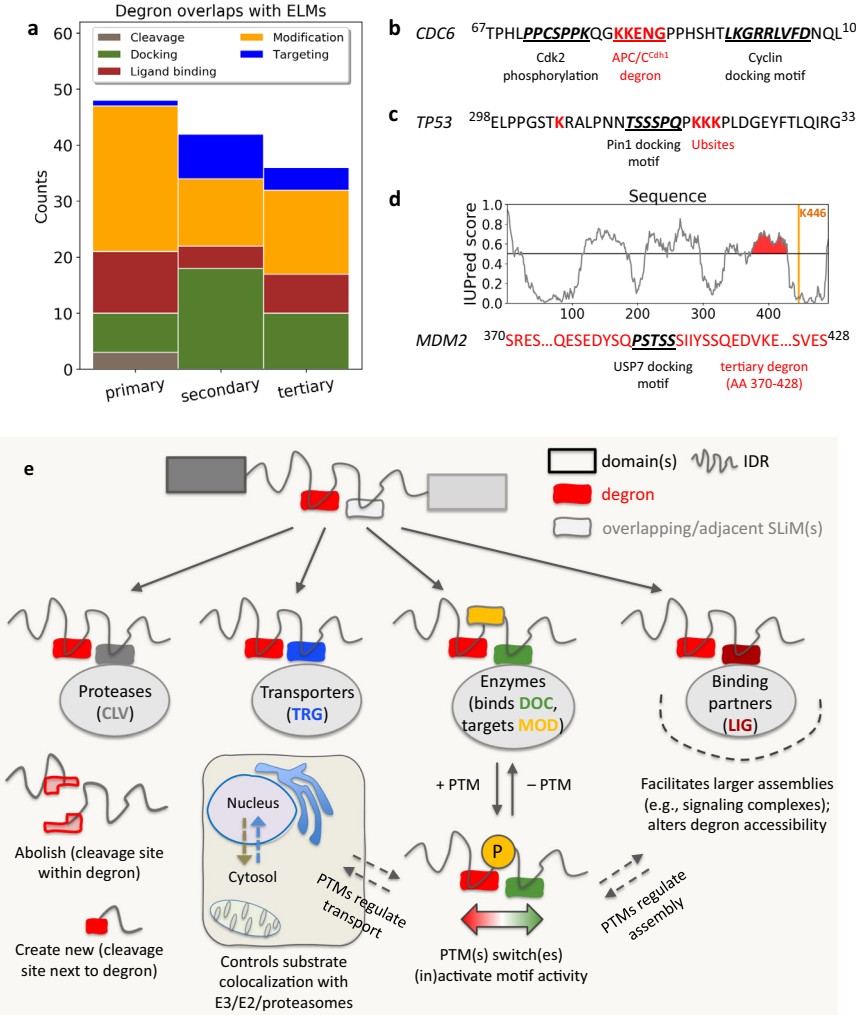

**Fig. 3 Overlap of degrons with Eukaryotic Linear Motifs (ELMs). a** Plot showing the number of overlapping (or adjacent) ELMs relative to the primary, secondary and tertiary degrons in our dataset. The overlapping ELMs are color coded according to their functional category (defined by ELM curators). **b–d** Examples from each degron category (degron sequence in red) showing the details of one (or more) overlapping/adjacent motifs (shown in bold italics). Additionally, panel (**d**) shows the IUPred[75] predicted disorder profile of the substrate (MDM2) showing how the tertiary degron (the region shaded red in the disorder profile) was defined as the IDR nearest to the degradation-linked Ubsite, K446. **e** Outline demonstrating multiple possibilities of degron masking based on the different ELM types and their functional outcomes.

Ubsites and tertiary degrons. Although the majority of overlapping modification motifs were related to phosphorylation, we also found overlapping SUMOylation motifs. For Ubsites, crosstalk with sumoylation can be important for degradation regulation[40]. Furthermore, since many degradation-linked Ubsites are actually located within the tertiary degron[14], sumoylation could also regulate proteasomal degradation initiation. Similarly, in addition to kinases, other enzymes also regulate degradation. For example, p53 has a docking motif for the peptidyl-prolyl cis-trans isomerase, Pin1, located next to several Ubsites (Fig. 3c). High Pin1 activity results in weak p53 ubiquitination[41]; in contrast, when Pin1 activity is low, strong p53 polyubiquitination and increased degradation was observed[41]. Upon DNA damage, p53 associates with Pin1 and is stabilized. Thus, it is plausible that due to the adjacency of Ubsites and the Pin1 docking site, when Pin1 levels and activity are high, it not only catalyzes cis-trans isomerization, but also effectively masks the Ubsites, preventing p53 polyubiquitination. Another example highlights a deubiquitinase (Usp7) docking motif that negatively regulates Mdm2 degradation[42]. Lys446 is the major site for Mdm2 auto-ubiquitination[43] and the tertiary

degron, containing the Usp7 docking motif, is located just upstream (AA 370–428) (Fig. 3d). Therefore, Usp7 likely stabilizes Mdm2 via a dual mechanism: it deubiquitinates Lys446 and sterically hinders proteasomal access to the tertiary degron (effectively weakening affinity to and decreasing proteasomal residence time).

Ligand binding motifs were also found to overlap degrons (Fig. 3a): these recruit diverse partners in order to assemble macromolecular complexes, such as signaling assemblies[34]. Incorporation into complexes would stabilize the substrate by masking its degrons and thereby determine its functional lifetime (Fig. 3e). This would also serve to degrade 'excess' free subunits, regulate complex stoichiometry and prevent non-functional interactions[44]. Cellular compartmentalization is another important factor regulating spatiotemporal substrate availability: an overlapping targeting motif (such as a nuclear localization signal or nuclear export signal) could alter the localization of substrates vis-à-vis cognate E3s (Fig. 3e). Localization is also often controlled by phosphorylation[45], meaning that modification motif(s) typically form part of the regulatory module. Finally, the overlap of a cleavage motif with an internal degron can also

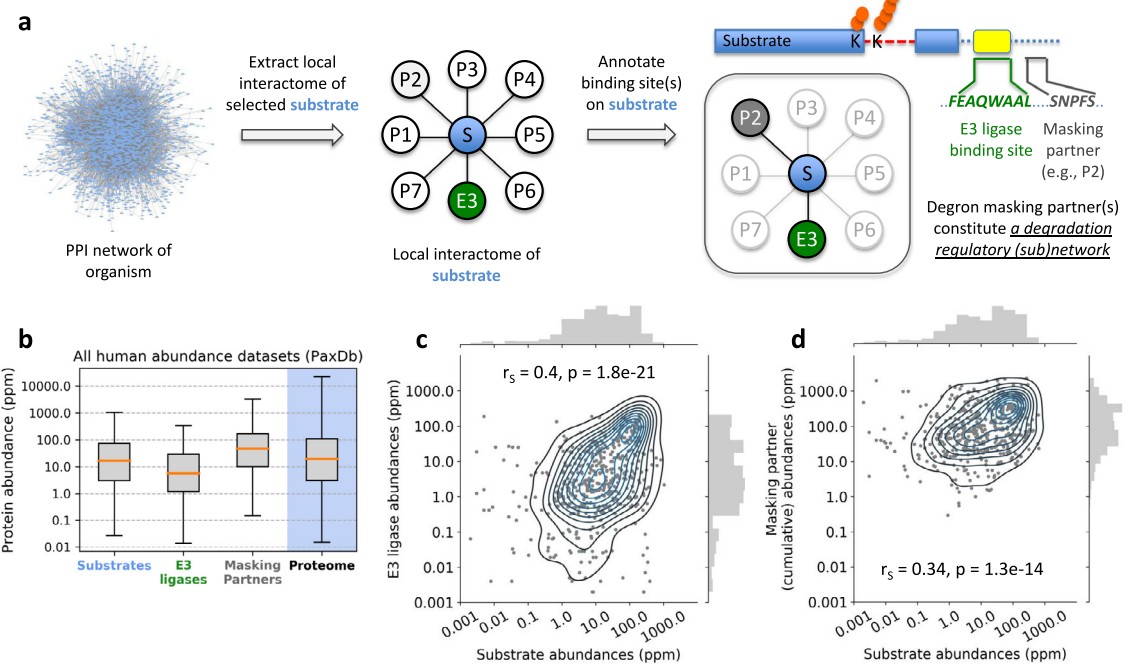

**Fig. 4 Substrate degradation regulatory modules and their analysis based on protein abundances. a** After identifying interacting proteins for a selected substrate (its "local interactome"), available binding site/motif information is used to identify partners whose binding sites overlap substrate degrons. This subset of partners constitutes a degradation regulatory module (or subnetwork) for the substrate of interest. **b** Based on this concept, we identified the protein components that comprised the primary degron regulatory subnetworks for selected human substrates (Table 1). Relative protein abundances were compared after grouping into the three relevant categories: substrates, E3 ligases (including E3 adaptor subunits), and the degron masking alternate partners (APs). The total number of abundance data points for each of these groups: substrates ($N = 1069$), E3s (472) and APs (3838). For comparison, the abundance distribution for the entire human proteome is also shown. Outliers not shown in the boxplots. **c** Scatter plot of substrate and corresponding E3 ligase pairwise abundances across all 170 PaxDb datasets. Abundances were taken from each individual PaxDb dataset, whenever data for both proteins of a pair (i.e., substrate, E3) were available. **d** Scatter plot between the abundance value of each substrate and summed abundances of all its corresponding masking partners (APs), from each PaxDb dataset, wherever data for all the proteins of each group (i.e., substrate and its corresponding APs) were available. Spearman's correlation coefficients ($r_S$) and corresponding *p*-values are shown on the figures.

result in several scenarios (Fig. 3e): it could either abolish an existing degron, if the cleavage site were located within the degron; or activate it if cleavage makes it more accessible. Furthermore, cleavage by endoproteases can potentially form new N- or C-degrons[46]. In summary, these diverse scenarios utilize multiple, distinct, sequential or mutually exclusive PPIs to exert conditional control over degradation. Again, by embedding degrons within functionally dense IDRs, sets of overlapping motifs can form complex degradation regulatory modules (Fig. 3e).

**Protein abundances and binding competition in degradation-regulatory modules**. Degron overlapping motifs combinatorially create degradation regulatory modules (Fig. 4a). We analyzed primary degron overlapping ELMs of human substrates (Supplementary Data 6), by annotating the partner proteins, APs, recruited to each overlapping motif, based on database and manual curation (Table 1). Several of them were modifying enzymes (e.g., kinases) that affect substrate levels by modulating the PTM status of the degron or its neighborhood, changing interaction specificity towards different partner(s). To assess binding competition and to estimate relative motif occupancies, we analyzed cellular abundances of the protein groups (substrates, E3s, and APs). Competition should be predominantly driven by relative abundances since the motif lengths are comparable and motif-based interactions are mostly weak and transient ($K_d$ typically in the low μM range)[17]. We also collected available binding ($K_d$) data of substrate-E3 and substrate-AP

pairs for primary degron containing substrates (Supplementary Data 9). Overall, the $K_d$ distributions of these two groups did not show any significant difference (Supplementary Fig. 7a), thereby indicating that relative protein abundances should indeed be more of an influencing factor in competitive binding to overlapping motifs.

We obtained protein concentrations from PaxDb[47], which provides normalized, proteome-wide abundances (see Methods). Overall, E3s were significantly lower in abundance compared to substrates ($P = 1.25\text{E}{-}12$), and, more strikingly, E3s were ~10-fold less abundant than APs ($P = 1.13\text{E}{-}62$) (Fig. 4b). PaxDb datasets are grouped into four categories (whole-organism, integrated, tissues and cell lines; Supplementary Data 10). Quantitatively similar patterns of relative abundances (E3s < substrates < APs) emerged even when segregated by dataset type (Supplementary Figs. 8, 9). On average, cellular concentrations of E3s are limiting relative to APs; therefore, APs should exert strong regulatory effects by masking degrons for a substantial fraction of substrate lifetime. For those substrates where we obtained $K_d$ data for both substrate-E3 and substrate-AP(s) binding, the measured ranges of $K_d$ values for AP interactions are either very similar to or smaller than the $K_d$'s measured for E3 interactions (except for p53) (Supplementary Fig. 7b). Taken in conjunction with the greater abundances of APs as compared to E3s, this gives greater confidence to the proposed model that APs will mask substrate degrons for a significant proportion of substrate lifetime. Moreover, the formation of alternative complexes ensures functional lifetime of the substrates and enables their cellular functions, via competitive and

**Table 1 Selected human degradation substrates, their corresponding E3 ligases (or the adaptor subunit, in case of multi-subunit E3s) and degronon components.**

| Substrate protein (Gene name, UniProt ID) | Primary degron | E3 ligase (Gene name, UniProt ID) | Degronon components[a](Gene name, UniProt ID) |
|---|---|---|---|
| G1/S-specific cyclin-E1 (CCNE1, P24864) | [393]LLTPPQS[399] | FBXW7 (Q969H0) | GSK3B (P49841), CDK2 (P24941) |
| Numb-like protein (NUMBL, Q9Y6R0) | [577]FEAQWAAL[584] | MDM2 (Q00987) | EPS15 (P42566), EPS15L1 (Q9UBC2) |
| Cellular tumor antigen p53 (TP53, P04637) | [19]FSDLWKLL[26] | MDM2 (Q00987) | PIN1 (Q13526), CSNK1D (P48730), GSK3B (P49841), PRKDC (P78527), ATM (Q13315), [b]CREBBP (Q92793) |
| Myc proto-oncogene protein (MYC, P01106) | [55]LLPTPPLS[62] | FBXW7 (Q969H0) | PIN1 (Q13526), BIN1 (O00499), GSK3B (P49841), MAPK1 (P28482), [b]DYRK2 (Q92630), [b]PPP2R5D (Q14738) |
| Mitotic checkpoint serine/ threonine-protein kinase BUB1 (BUB1, O43683) | [534]NKENY[538] | FZR1 (Q9UM11) | CDC20 (Q12834) |
| Claspin (CLSPN, Q9HAW4) | [29]DSGQGS[34] | BTRC (Q9Y297) | PLK1 (P53350), CASP3 (P42574) |
| Cyclin-dependent kinase inhibitor 1B (CDKN1B, P46527) | [183]SVEQTPKK[190] | SKP2 (Q13309), CKS1B (P61024) | CCNE1 (P24864), CDK2 (P24941), PIN1 (Q13526), YWHAQ (P27348), [b]AKT1 (P31749) |
| Cyclin-dependent kinase inhibitor 1 (CDKN1A, P38936) | [145]TSMTDFYHSKRRL[157] | DTL (Q9NZJ0) | PCNA (P12004), AKT1 (P31749), PRKCA (P17252), KPNA1 (P52294), [b]CCNE1 (P24864), [b]CDK2 (P24941) |
| Uracil-DNA glycosylase (UNG, P13051) | [58]PGTPPSS[64] | FBXW7 (Q969H0) | [b]GSK3B (P49841), RPA2 (P15927), [b]CCNE1 (P24864), [b]CDK2 (P24941) |
| Transcription factor AP-1 (JUN, P05412) | [227]EEPQTVPEM[235] [236]PGETPPLS[243] | COP1 (Q8NHY2), FBXW7 (Q969H0) | GSK3B (P49841), DYRK2 (Q92630), PRKDC (P78527), UBE2I (P63279), SUMO1 (P63165) |
| Cell division control protein 6 homolog (CDC6, Q99741) | [80]KKENG[84] | FZR1 (Q9UM11) | CCNE1 (P24864), CDK2 (P24941) |
| FYVE, RhoGEF and PH domain-containing protein 3 (FGD3, Q5JSP0) | [75]DSGIDS[80] | BTRC (Q9Y297) | GSK3B (P49841) |
| Catenin beta-1 (CTNNB1, P35222) | [32]DSGIHS[37] | BTRC (Q9Y297) | GSK3B (P49841), [b]CSNK1A1 (P48729), [b]AXIN1 (O15169) |
| FYVE, RhoGEF and PH domain-containing protein 1 (FGD1, P98174) | [282]DSGIDS[287] | BTRC (Q9Y297) | GSK3B (P49841) |
| Sequestosome-1 (SQSTM1, Q13501) | [347]DPSTGE[352] | KEAP1 (Q14145) | GABARAP (O95166), GABARAPL1 (Q9H0R8), MAP1LC3A (Q9H492) |
| Cyclin-dependent kinase inhibitor 1 C (CDKN1C, P49918) | [306]SVEQTPRK[313] | SKP2 (Q13309), CKS1B (P61024) | CDK2 (P24941), [b]CCNE2 (O96020) |
| Zinc finger protein SNAI1 (SNAI1, O95863) | [95]DSGKGS[100] | BTRC (Q9Y297) | GSK3B (P49841) |
| Hypoxia-inducible factor 1-alpha (HIF1A, Q16665) | [400]LAPAAGDTIISLDF[413] | VHL (P40337) | EGLN1 (Q9GZT9), UBE2I (P63279), SUMO1 (P63165) |
| Double-strand-break repair protein rad21 homolog (RAD21, O60216) | [169]EIMREG[174] (internal N-end degron exposed by cleavage after R172) | UBR1 (Q8IWV7) | [b]ATE1 (O95260), ESPL1 (Q14674) |
| Sterol regulatory element-binding protein 1 (SREBF1, P36956) | [425]LTPPPS[430] | FBXW7 (Q969H0) | GSK3B (P49841), PLK1 (P53350), [b]CDK1 (P06493), [b]CCNB1 (P14635) |

[a]Primary degron overlapping motifs were identified from the ELM database (annotated in Supplementary Data 6). The specific proteins that bind to these overlapping motifs were identified based on descriptions provided in ELM[35], switches.ELM[78] and UniProtKB[33].
[b]Added as degronon components, based on literature description, as being involved in affecting the stability of the corresponding substrate by binding to (and/or involved in post-translationally modifying) primary degron overlapping sites.

context-dependent degron masking. Thus, degradation is tightly regulated, with defined spatiotemporal windows when substrate degrons get unmasked. Once triggered, given the processive nature of E3s, their low abundances should suffice for efficient degradation. Furthermore, additional mechanisms can directly regulate E3 activity[48], including PTM-mediated (in)activation and/or changes in cellular localization.

Next, we analyzed substrate–E3 and substrate–AP abundance correlations. Although we expected large (biological) variability since the analysis included all PaxDb datasets, substrate–E3 pairs exhibited a moderate, but statistically significant correlation (Spearman $r_S = 0.4$; $P = 1.8E-21$) (Fig. 4c). Balanced abundances of interacting proteins can prevent promiscuous (potentially deleterious) interactions[49], which is particularly important here, since most of the substrates were IDR-rich, interaction-prone hubs. IDR-containing proteins have tightly regulated endogenous levels[50] with targeted degradation constituting an important avenue for regulation. Therefore, there should be a strong pressure to maintain correlated E3–substrate abundances across diverse cell types (Fig. 4c). In contrast, substrate–AP pairs showed

a weak correlation ($r_S = 0.2$) (Supplementary Fig. 10), likely arising from factors such as context-dependent signaling in different cell types. Moreover, multiple APs compete for binding to each substrate (Table 1). Unlike substrate–E3 relationships, where a tighter connection in terms of abundances is required for the control of substrate function (e.g., the duration and amplitude of activity via degradation), the situation with APs is more complex. However, in contrast to individual APs, summed AP abundances per substrate showed a larger, significant correlation ($r_S = 0.34$; $P = 1.3E-14$) (Fig. 4d), indicating that higher (cumulative) AP abundances translate into higher substrate levels, likely reflecting greater stabilization attributable to degron masking.

**Biological and disease-linked variation within degradation regulatory modules.** PaxDb provides abundance datasets for normal human tissues as well as cell lines; among the latter, the majority were cancer cell lines (Supplementary Data 10). We used this data to investigate how abundances of degradation module components (Table 1) vary within and between tissues and cell lines. Many proteins showed large variations within each category (Fig. 5a and Supplementary Fig. 11), demonstrating that degradation regulatory modules may be configured differently (in terms of component abundances and therefore relative motif occupancies), depending on cellular (or disease) context. Moreover, when compared between tissues and cell lines, several oncogenic substrates (e.g., Bub1, Claspin, Myc, p53; Fig. 5a), E3s and APs showed significant differences (also see Supplementary Discussion).

We propose that by simultaneously exploring abundance variations of all proteins within each degradation regulatory module (e.g., the p53 module, Fig. 5b), one can potentially identify relevant (e.g., disease-specific) changes in the expression of stabilizing (or degradative) complexes. Fig. 5c is a heatmap showing the abundance variation of the p53 degradation regulatory module components (highlighted in Fig. 5b on the network view) across a selection of PaxDb datasets. The left-most group of datasets in Fig. 5c are "integrated" datasets, generated by PaxDb following a weighted averaging procedure for organisms/ tissues for which multiple individual experimental datasets were available, leading to a "best-estimate" abundance quantification[47]. The color scale uses ranked abundances indicating the relative position of each protein within each abundance dataset. Rank 1 and 100 indicates that the specific protein occurs within the top 1% (most abundant) and bottom 1% (least abundant), respectively, of that dataset (see Methods). p53 abundances were very low in normal (unstressed) tissues and no data were available for almost all tissue datasets (the respective cells are therefore colored white indicating absent abundance data; it is very likely that wt-p53 levels were below measurement threshold) (Fig. 5c). In contrast, p53 abundances increase in all the cell lines shown (all are cancer cells, except HEK293) and therefore become measurable. It is possible that (much of) the p53 species measured in these cells could be oncogenic, gain-of-function (GOF) p53 missense mutants (mut-p53) that impart proliferative properties to cancer cells[51]. Stabilization may be a result of the upregulation of APs or, alternately, oncogenic mutations may directly increase protein stability or affect the binding of degron-shielding partners. In fact, mut-p53 is stabilized under tumor-related stress[51] and in conjunction with higher CBP abundances (Fig. 5c), could imply that many GOF properties of mut-p53 may result from the higher amounts of mut-p53–CBP/p300 complexes that can transactivate tumor cell gene expression. Indeed, inhibiting the mut-p53-p300 interaction abolished tumor-promoting properties of mut-p53[52]. Expression of several kinases that phosphorylate and stabilize p53 upon stress (ATM, DNA-PK, etc.) are also

higher in cell lines (Fig. 5c). ATM phosphorylates Ser15 (overlapping the Mdm2-binding degron; Fig. 5b) and stabilizes mut-p53 by preventing polyubiquitination; ATM inhibition restores mut-p53 polyubiquitination[53].

Therefore, the approach we propose underlines the importance of studying the relative expression of alternate complexes within each substrate's degradation regulatory network (defined on the basis of degron overlapping PPI sites; Figs. 4a, 5b). For a substrate of interest and its local interactome (Fig. 5b), by identifying specific interaction partners whose binding sites overlap and thereby shield degrons (Fig. 5b), we can use that PPI subset for further correlative analysis of gene/protein expression (demonstrated here using protein abundances, Fig. 5c). Whenever such information is available, we can deduce how altered abundances (Fig. 5c) shift the balance of competition, highlighting the (sub) set of complexes with higher expression likelihood in specific cell (or disease) subtypes. These could then be prioritized for functional analysis of downstream signaling and potentially targeted for therapy.

**Co-regulated degradation modules ("degronons") point to functional assemblies.** The modular organization of the proteome requires the co-regulation of functional units (complexes, pathways, networks) at all levels: transcription, translation and protein degradation. Such coordination clearly exists for protein production[1] and pertinent concepts, such as operon and regulon, have been well established[54]. In this section, we analyzed PPI networks of the yeast *Saccharomyces cerevisiae* to understand globally, how physical interactions coordinate protein stabilities and functional relationships, and specifically, if co-regulated degradation units ("degronons") can be identified. Yeast is an extensively studied model organism with multiple high-coverage, high-quality datasets (protein half-lives, abundances, gene co-expression, GO annotations) available. We selected two PPI networks (see Methods): (1) a high-confidence network of *S. cerevisiae* soluble proteins ("Collins" network[55]), and, (2) the multi-validated BioGRID yeast network[56].

Each network was decomposed into binary PPI pairs (forming direct, physical interactions) and their relevant properties were analyzed (Fig. 6a). Firstly, we observed a marked preference for interacting pairs to exhibit highly similar half-lives (Fig. 6b); significantly different in behavior from random PPI networks (see Methods). Importantly, the pattern persisted for both networks and multiple half-life datasets (Supplementary Fig. 12), reflecting a biological constraint for physically interacting proteins to have correlated stabilities. This could be interpreted as active (co-) degron masking and/or physical co-stabilization due to binding.

These correlated local interactome elements can be considered as "elementary degronons", representing synchronized protein pairs. To analyze the extent to which they correspond to functional units, we grouped all binary PPI pairs into three categories, based on their half-life ratios (calculated as smaller/ larger $T_{1/2}$ value): resulting in pairs with similar (0.8–1.0), different (0.5–0.8) and very different (0.0–0.5) half-lives. The number of pairs in the 'similar' category were much higher than in the other categories (Fig. 6b). For each protein pair, we analyzed their functional similarity (calculated as GO Biological Process semantic similarity; see Methods) (Fig. 6c) and gene co-expression (quantified as the Pearson correlation coefficient of their corresponding mRNA expression profiles; see Methods) (Fig. 6d). We observed highly significant increases of functional similarity and gene co-expression with increasing half-life similarity of the PPI pairs, indicating that functional protein modules (interacting pairs with correlated functions) are not only strongly co-regulated at the expression (mRNA) level but also

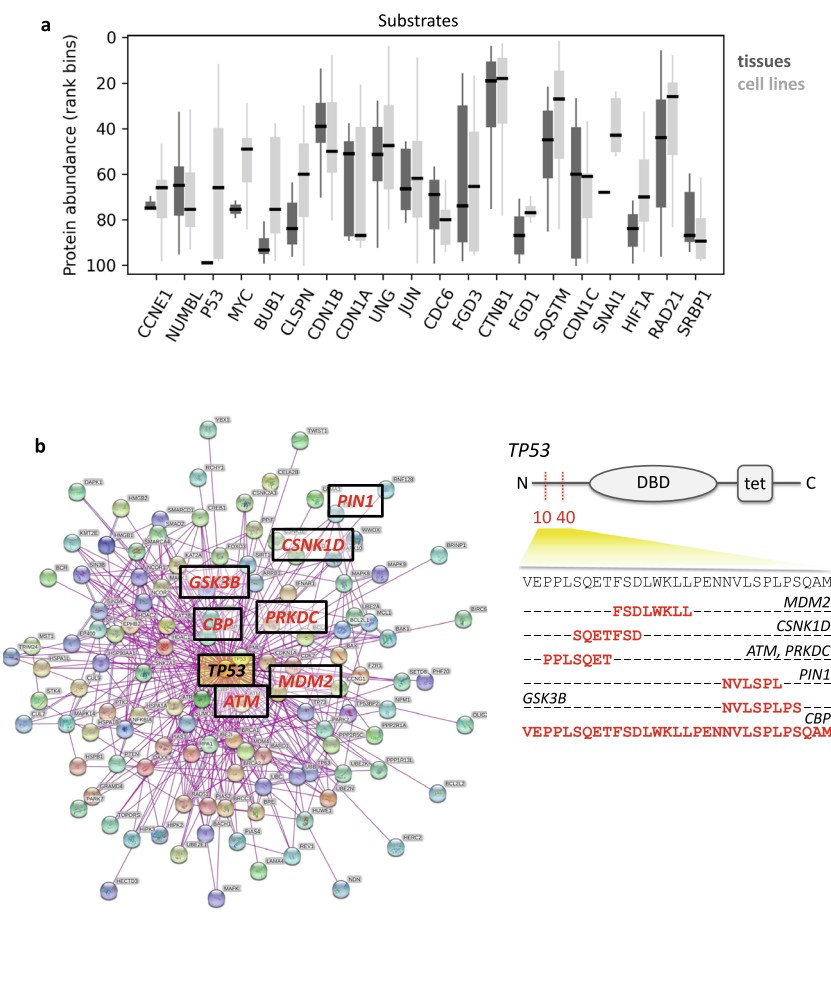

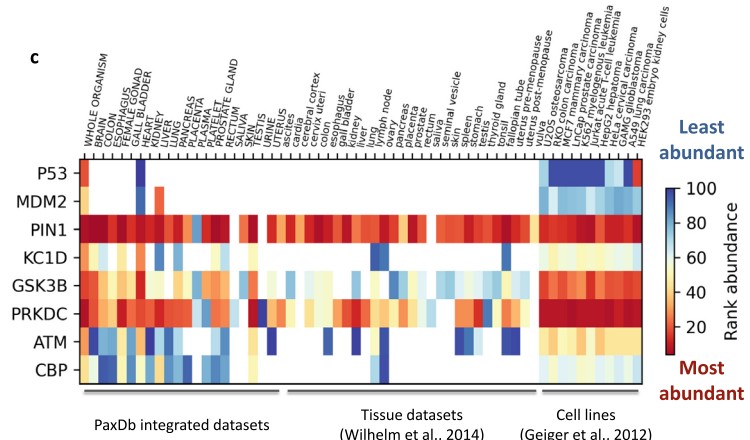

show high synergy in turnover. These observations were independent of the PPI network, as both the Collins and BioGRID networks showed similar profiles. In contrast, the relative abundance of PPI pairs (the ratio of abundances of the two proteins comprising each interacting protein pair) showed a less dramatic increase as a function of the $T_{1/2}$ ratio category (Fig. 6e), suggesting that subunit stoichiometry may be largely independent of component half-lives.

Having observed that direct partners in PPI networks exhibit a significant tendency towards similar half-lives (Fig. 6b), we next

asked whether this trend extended to non-adjacent pairs along defined PPI paths, that would indicate larger, co-degrading functional units (degronons). To this end, we identified shortest paths between all (i,j) protein nodes in the Collins network (see Methods), and calculated $T_{1/2}$ ratios for indirectly linked pairs (i.e., 2nd, 3rd, …, nth neighbors). Moving along these paths, the strong $T_{1/2}$ ratio signal, seen for direct neighbors, dropped significantly as from the 3rd neighbor onwards, to levels observed for random PPI pairs (Fig. 6f). However, we reasoned that strong half-life correlations could persist over a longer range for

**Fig. 5 Protein abundance variations within degradation regulatory modules. a** Abundance variations of substrates (from Table 1) across the entire available set of tissue and cell-line PaxDb datasets. Number of available protein abundance data points per substrate (tissue, cell line data points): CCNE1 (3,17), NUMBL (26, 36), P53 (1, 23), MYC (2, 16), BUB1 (18, 36), CLSPN (6, 35), CDN1B (29, 18), CDN1A (5, 19), UNG (30, 36), JUN (6, 20), CDC6 (3, 26), FGD3 (12, 18), CTNB1 (73, 52), FGD1 (13, 27), SQSTM (43, 52), CDN1C (18, 24), SNAI1 (1, 5), HIF1A (3,7), RAD21 (42,52) and SRBP1 (12, 24). **b** Map of the interaction partners of human p53 (TP53). Partners that mask the MDM2-binding primary degron of p53 are highlighted (within boxes) in the interactome map. All these proteins bind to the same segment of p53 (aa 10–40, shaded yellow) as the E3 ligase Mdm2 and the specific binding motifs for each of these partners are marked in red, and shown below the domain diagram of p53. **c** Abundance variations for components of the p53 primary degron's degradation regulatory network are shown as a heat map across a selected subset of PaxDb datasets ("integrated" datasets, individual tissue datasets derived from Wilhelm et al.[76] and cell line datasets derived from Geiger et al.[77]). Protein abundances are represented as ranked abundances that signify the relative abundance of each protein within an abundance dataset (see Methods). Darker red color indicates that the protein is among the most abundant ones within the given dataset, whereas dark blue indicates that it is among the least abundant ones (each column is from a specific abundance dataset). Cells colored white indicate missing abundance data for that protein in that abundance dataset.

pathways whose members had close functional similarity. To validate this possibility, we identified the subset of pathways where every member had high BP GO SemSim to the first (starting) member. Within such functionally connected paths, every protein exhibited highly similar $T_{1/2}$ values to the first member (Fig. 6g). Remarkably, correlated stabilities extended as far as 6th (or farther) neighbors, satisfying the criterion for degronons, coordinated degradation sub-networks within the interactome. For degronons of length ≥6, we identified protein members and their participating complexes (Supplementary Data 11): their functions included translation initiation (e.g., eukaryotic translation initiation factor, eIF1, 2, and 3 complexes; Fig. 6h) and RNA splicing (U2 and U6 snRNP, U4/U6.U5 tri-snRNP complexes and splicing factor, SF3 components of the spliceosome). These examples make a strong case for the existence of interconnected, functional modules whose degradation may be synchronized by degron masking.

Finally, although it might seem conceivable that any (even non-specific) interaction might physically stabilize proteins, all the data on degron masking presented in this paper collectively argue that stabilization mediated by PPIs is necessarily specific and regulated. Fig. 6i presents additional data showing a complete absence of correlation between protein half-lives and the (total) number of interaction partners, or between half-lives and the mean abundance of all partners (Fig. 6i, inset), indicating that only certain interactions with specific degron-masking partners should impart stabilization in a biological context.

## Discussion

Protein turnover receives dynamic input from multiple pathways, manifesting in degron masking by alternate PPIs that integrate cell state and signaling information. In vivo dynamics of complexes in which degrons get shielded determines the functional longevity of proteins. Complex assembly affects degron exposure and activity either by direct masking within the interface (Fig. 1) or by inducing conformational changes leading to intramolecular shielding. Here, we presented a comprehensive analysis of degron-overlapping binding sites (Fig. 2) and sequence motifs (Fig. 3), outlining a broad mechanistic framework of stability regulation by degron masking. We expect that the framework presented here, combining degron identification and partner binding site information, will inspire dedicated studies on substrates of interest and provide a foundation for guiding experimental design.

Given that degrons are mostly present in IDRs[14,15], both local dynamics and long-range structural rearrangements upon binding could contribute to masking. Locally, folding-upon-binding of IDRs influences degron accessibility: for example, free IκBα has a C-terminal IDR responsible for its Ub-independent, 20 S proteasomal degradation. However, in complex with NF-κB, the IDR gets folded and masked in the interface, stabilizing IκBα[57]. Local

conformational preferences are also influenced by PTMs[58], which can impact degron accessibility. IDRs host multiple PTM sites, allowing interplay between modifying enzymes and regulatory partners (Fig. 3). Furthermore, PTMs can mask degrons (stereo) chemically by modifying the charge, size and topological features of binding sites. For example, ubiquitination itself can modulate stability by regulating PPIs[59]. Importantly, IDRs also enable Ub-independent, degradation "by default" via 20 S proteasomes[60] and the masking of IDRs by PPIs are known to inhibit degradation[60,61]. Since IDR-containing proteins may be subject to both Ub-dependent and Ub-independent pathways, the fact that IDRs are enriched in PPI sites (Figs. 2c, 3a) strongly suggests that IDR masking could regulate both 20 S and 26 S proteasomal degradation pathways.

In the context of the tripartite degron model[14], the E3 binding motif (primary degron) is clearly important for Ub-dependent degradation, conferring specificity to substrate recruitment[13] and making primary degron masking an important regulatory step. However, Ubsites and degradation initiation segments are equally important for the complete degradation cycle. For example, Ubsite mutations block substrate degradation leading to disease[14]. Masking substrate Ubsite(s) will therefore make them inaccessible (or less accessible) for E2-catalyzed Ub-transfer and can stabilize substrates, independent of primary degron recognition. Furthermore, the dynamics of Ubsite-masking complexes could result in differences in the rates of synthesis of polyUb chains on substrates. That could hypothetically have interesting implications, such as influencing the ordering of substrate degradation in cell cycle regulation by the anaphase-promoting complex/cyclosome[62] or cell-type specific regulation of protein stability. Similarly, if the tertiary degron (disordered proteasomal entry site) is bound by a partner (Fig. 2f), this results in substrate stabilization[32], demonstrating that the tertiary degron can also be independently regulated via masking. Another example involves the efficient degradation of Calpain-cleaved Retinoblastoma protein, involving tertiary degron exposure following internal cleavage (the tertiary degron in this case can be considered as shielded intra-molecularly)[63]. In summary, the masking of each of the tripartite degron elements (both by intra and inter-molecular PPIs) may independently regulate degradation, with combinatorial masking of multiple substrate degrons also possible (Supplementary Figs. 4, 5).

In this study, we also proposed a framework for defining degradation-regulatory PPI modules (subnetworks) based on degron-overlapping binding sites (Fig. 4) and analyzed their tissue-specific and disease-linked variation based on component abundances (Fig. 5). Given that alternate, overlapping binding motifs that recruit competing partners are equivalent in length and a comparison of $K_d$ values showed that no significant differences exist between E3 and APs that bind to these overlapping motifs (Supplementary Fig. 7), binding occupancy would depend

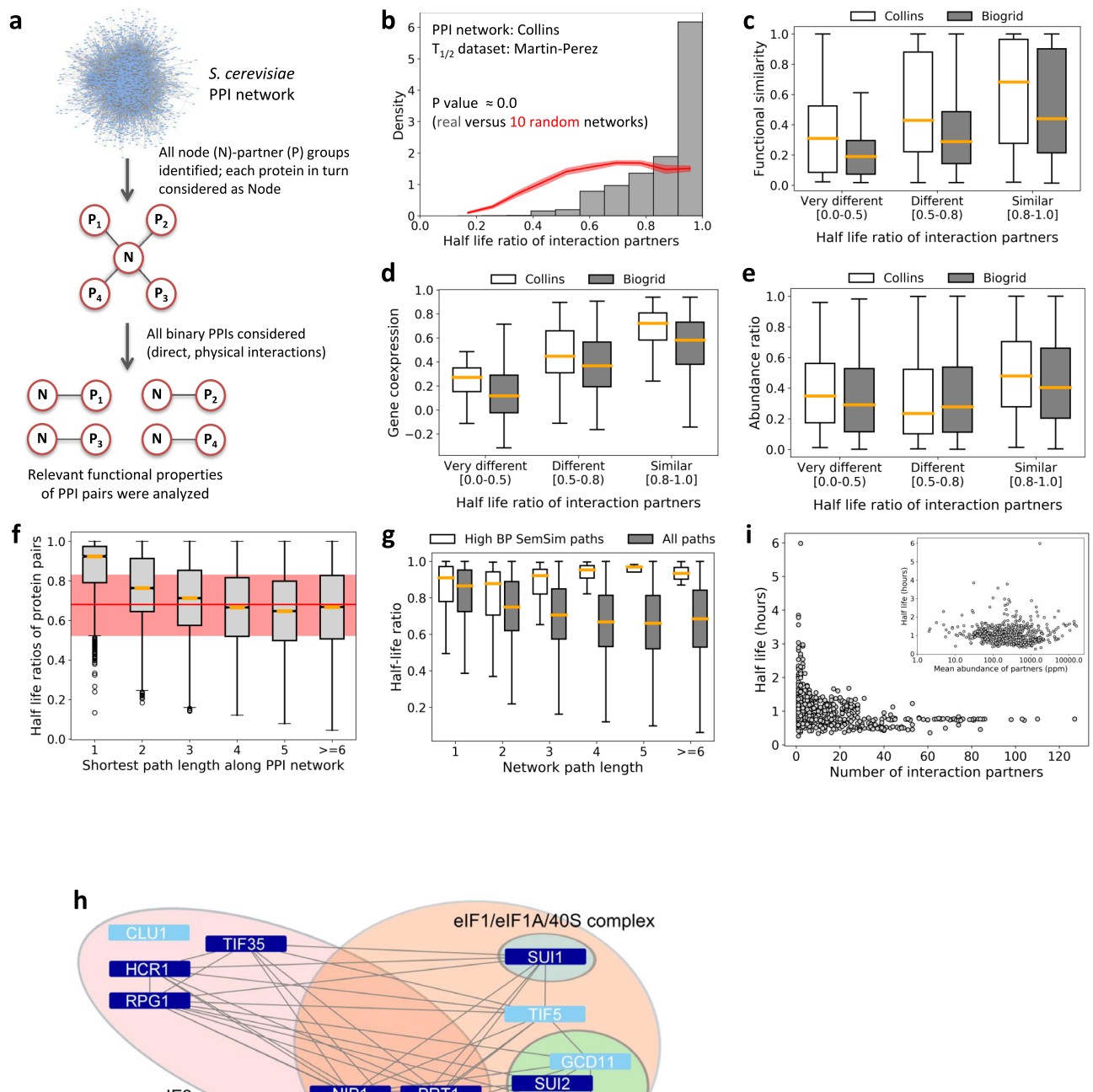

largely on relative abundances of the proteins involved. The formation of substrate-AP complexes should also modulate substrate residence times on cognate E3 ligases and via this mechanism disassociate E3 abundances as a limiting factor for substrate turnover, i.e., once degrons get unmasked, E3 activity might no longer be constrained by its typically low abundances (Fig. 4b). It should also be pointed out that beyond $K_d$ and abundances, interaction kinetics also will influence the balance of competing interactions. However, such data are extremely scarce and, being highly sensitive to experimental conditions, are perhaps less reliable than thermodynamic values. Here, for the abundance analysis of degradation regulatory networks, we focused on PPI partners that masked primary degrons (Figs. 4, 5),

whereas the full network would include partners that mask secondary and tertiary degrons as well. Such detailed studies, of future interest, would necessitate focusing on select substrates, employing structural, biochemical and cellular approaches. For instance, identifying degrons and characterizing a (more) complete set of degron masking partners is challenging in itself. Manifestly clear, even from this initial analysis, is the inherent complexity of such networks, both in terms of the number of potential regulatory partners and their cell-specific variation (Fig. 5), resulting in diverse functional outcomes. Detailed analysis of such systems is beyond the scope of this study and further work will be required to identify alternate complexes and their relative expression associated with disease.

**Fig. 6 Interactome-level stability and functional analysis of the *S. cerevisiae* proteome. a** Dissection of the yeast PPI network into binary, physically interacting protein pairs for the analysis of relevant pairwise properties. **b** Distribution of half-life ratios of PPI pairs compared versus those of PPI pairs from randomized networks (average ± 1 standard deviation range, calculated on 10 random networks, shown in red). **c** Pairwise functional similarity (estimated using GO BP semantic similarity score) for PPI pairs, grouped according to their half-life ratios, from the Collins and BioGRID networks. The numbers of PPI pairs in the three half-life ratio groups (Very different, Different and Similar) were respectively: 25, 250, 730 (Collins) and 104, 532 and 968 (Biogrid). *P*-values between the groups: Collins (Very different vs. Different: 0.014, Very different vs. Similar: 8.1E-5, Different vs. Similar: 1.3E-7); Biogrid (Very different vs. Different: 5.9E-6, Very different vs. Similar: 2.3E-16, Different vs. Similar: 3.9E-16). Half-life dataset from Martin-Perez was used, as in (**b**). **d** Gene co-expression values for corresponding PPI pairs, grouped according to their half-life ratios, from the Collins and BioGRID networks. The numbers of PPI pairs in the three half-life ratio groups (Very different, Different and Similar) were respectively: 71, 830, 2422 (Collins) and 250, 1332 and 2047 (Biogrid). *P*-values between the groups: Collins (Very different vs. Different: 1.08E-13, Very different vs. Similar: 4.0E-31, Different vs. Similar: 2.7E-104); Biogrid (Very different vs. Different: 1.4E-39, Very different vs. Similar: 4.8E-88, Different vs. Similar: 3.5E-78). Half-life dataset from Martin-Perez was used for grouping the PPI pairs. **e** Protein abundance ratios for PPI pairs, grouped according to their half-life ratios, from the Collins and BioGRID networks. The numbers of PPI pairs in the three half-life ratio groups (Very different, Different and Similar) were respectively: 75, 942, 2844 (Collins) and 282, 1498 and 2295 (Biogrid). *P*-values between the groups: Collins (Very different vs. Different: 0.0014, Very different vs. Similar: 0.001, Different vs. Similar: 1.0E-68); Biogrid (Very different vs. Different: 0.4, Very different vs. Similar: 9.4E-8, Different vs. Similar: 2.0E-25). Half-life dataset from Martin-Perez was used for grouping the PPI pairs. Outliers are not shown for the boxplots in panels (**c**–**e**). **f** Half-life ratios of all protein (node) pairs in the Collins network as a function of network distance (i.e., path length, based on shortest paths derived between each pair of nodes). The number of data points (i.e., half-life ratios) per path length category: 3861 (path length = 1), 13037 (2), 18859 (3), 24734 (4), 28607 (5) and 48146 (>=6). For comparison, the red line corresponds to the median half-life ratio for random PPI pairs ('direct' partners from random networks, i.e., path length = 1) while the red zone shows the 25th to 75th percentile range of the same random distribution. Half-lives were from the Martin–Perez dataset. **g** Half-life ratios of protein pairs along all shortest network paths (1st protein versus every pathway member) as a function of network (path) length in the Collins network. The subset of paths comprising proteins having high functional similarity ("High BP SemSim paths", defined using a BP SemSim cutoff of 0.6; i.e., each member had a BP SemSim value ≥0.6 compared to the first protein in the path) are compared to "All paths". The number of data points (i.e., half-life ratios) per path length bin: ("All paths" 1: 416572, 2: 421070, 3: 376141, 4: 311569, 5: 228675, >=6: 249655; "High BP SemSim paths" 1: 1577, 2: 473, 3: 111, 4: 28, 5: 4, >=6: 2). Outliers are not shown on the boxplot. **h** Examples of multiple, highly interconnected yeast degronon networks (see Supplementary Data 11 for the precise pathway definitions). These correspond to multi-protein complexes characterized by extremely high functional similarity and physical interconnectedness between their subunits (with multiple interaction edges connecting subunits and many shared subunits among the complexes; the proteins in light blue were not part of the degronon pathways but are members of the relevant complexes). Most importantly, they share very similar half-lives between subunits, central to the degronon concept. **i** Scatter plot of yeast protein half-lives as a function of their total number of interaction partners (from the Collins network) and the mean abundance of partners (inset).

Co-stabilization of interacting proteins appears to be a common regulatory paradigm, as reflected by the highly similar half-lives of yeast PPI pairs (Fig. 6b). Both large-scale[11] and individual studies have demonstrated that sequestration within complexes stabilizes subunits, and in certain cases, subunit stabilities were mutually interdependent (e.g., yeast Matα2/Mata1 and Drosophila Homothorax/Extradenticle; Supplementary Table 1). Such assembly principles regulate subunit stoichiometries since 'excess' proteins, not incorporated into functional complexes, are degraded to avoid potential negative effects[44]. This behavior points to likely degron unmasking upon subunit dissociation (and for unassembled subunits). Furthermore, in many co-regulated complexes, the loss of one subunit reduced the protein (but not mRNA) levels of the other subunit(s)[64]. Here, we observed that yeast PPI pairs having highly similar half-lives were: (1) functionally similar (Fig. 6c), and (2) exhibited strong mRNA co-expression (Fig. 6d). Thus, interacting proteins forming (parts of) functional complexes exhibited correlated expression as well as correlated stabilities (functional lifetimes). Significantly, even in larger multiprotein complexes, forming interaction 'paths' in PPI networks (Fig. 6g, h), the subunits possessed similar half-lives, indicating stringent requirement for collective stability to ensure functional coherence for the entire complex. Such interconnected groups of proteins, characterized by high functional coherence (e.g., co-complex membership), we termed 'degronons' to highlight their strongly correlated half-lives. We speculate that degron masking within such assemblies are likely to be a feature of their functional regulation. Another potential phenomenon to keep in mind would be the possibility of "collective destabilization", wherein a single E3 ligase potentially targets multiple substrates in the same regulatory network. However, it must be emphasized that degrons remain unidentified[65] for the majority of eukaryotic proteins, and therefore, our view on degron masking by PPIs is inevitably incomplete and will predictably expand in the future to include many more examples, eventually resulting in a more complete, proteome-wide view.

## Methods

**Degron datasets.** UPS degradation substrates and their experimentally validated degrons compiled in our previous study[14] were used (see Supplementary Data 1).

**PPI binding site information.** For each substrate, IntAct[24], UniProtKB[33] and the ELM resource[35] were queried for relevant binding site information (motif data in the case of ELM database) for partner proteins. Only experimentally identified PPI sites/motifs that overlapped with known degron(s) were considered. We considered strictly overlapping as well as adjacent binding sites/motifs; the latter defined as binding sites whose boundary was within 10 amino acids (along the sequence) of a known degron. IntAct data (in psimitab format) was parsed to keep only interactions between protein partners. We generated a distribution of their Interaction Confidence scores (Supplementary Fig. 6c); based on this, we retained PPI pairs with scores ≥0.3; this kept the large majority of data but removed a few pairs with low scores. From the "interactor features" annotation columns, we searched for the following information: "sufficient binding region", "binding-associated region" and "necessary binding region". Under these headings, IntAct annotates experimental information about the required binding site(s) for partner proteins, whenever that data is available. As shown in Supplementary Fig. 6a, the first two definitions accounted for the majority of PPI site information for our dataset. Of note, PPIs associated with such detailed annotations typically have higher confidence scores. Additionally, we also used mutation information falling under the following headings: "mutation decreasing interaction", "mutation disrupting interaction", "mutation disrupting interaction strength", "mutation decreasing interaction strength", "mutation disrupting interaction rate", "mutation increasing interaction", "mutation increasing interaction strength". If mutation(s) in/around a degron interferes with partner binding, that is also evidence that the relevant site is likely a physical interaction site. From UniProtKB, we manually curated the annotations for each substrate protein to identify relevant information that might indicate physical degron masking or information pointing to functional interference with degron function.

**Protein abundance data.** Protein abundances were obtained from The Protein Abundance Database (PaxDb)[47]. In total, 170 human abundance datasets were downloaded and analyzed. The datasets were grouped into the following four

categories: integrated datasets, whole organism datasets, tissue/organ-specific datasets and cell-line datasets (detailed list in Supplementary Data 10). These groups were based on the "#organ" and "#integrated" descriptor fields provided in the PaxDb data files. Since multiple whole-organism as well as tissue-specific datasets are available for *H. sapiens*, PaxDb employs a weighted averaging procedure to create "integrated" datasets which present "best-estimate" quantifications at the respective whole-organism (and specific tissue) levels[47]. PaxDb provides abundance data expressed in parts per million (ppm), describing relative abundances of each protein with reference to all the other protein molecules in the sample (i.e., the entire expressed proteome in that dataset). Therefore, within datasets, ppm values reflect proportional abundances and are useful for comparison-based statistics. Plots of dataset consistency, dataset coverage, size and other relevant properties are shown in Supplementary Fig. 13. PaxDb abundances are also comparable across cells of different volumes or across tissues of different cellular and extracellular compositions (see https://pax-db.org/help). Nevertheless, for cross-dataset comparisons, for example when comparing degradation regulatory module component abundances across cell and tissue types (Fig. 5 and Supplementary Fig. 11), we devised a ranking scheme to account for possible effects of dataset size differences on our observations. The rationale behind the ranking procedure was the following: PaxDb datasets span a certain range in terms of dataset size (i.e., number of detected proteins) and coverage (fraction of the proteome covered in each dataset); Supplementary Fig. 13c, d. However, since the ppm convention requires that the total abundance per dataset must always sum to ~1 million (Supplementary Fig. 13b), protein-specific comparisons of ppm values across datasets which have large differences in dataset size (or coverage) may not be ideal. For datasets of comparable size, this is much less of an issue. Therefore, to circumvent this potential problem, we converted ppm abundances in each dataset to ranked abundances. This was done by first sorting all the proteins in a given dataset (from highest to lowest by ppm value), then dividing the dataset into 100 equally populated bins and finally, calculating the ranked bin for each protein. The rank bin conversion was performed separately for each abundance dataset. A lower rank bin for a protein implies that it has higher abundance relative to the other proteins in the dataset. For example, rank 1 for a certain protein indicates that the protein is within the top 1% most abundant proteins measured in that dataset. This procedure essentially provides a numeric and directly comparable value that can be used for abundance comparisons both within and between datasets. However, we would point out that the essential trends in the data and their biological interpretations would remain the same irrespective of whether comparisons were made based on ppm abundances or ranked bins. For example, the same trends between relative abundances of substrates, E3 ligases and APs were clearly apparent when using either ppm values (Supplementary Fig. 8) or rank bins (Supplementary Fig. 9).

**Yeast protein–protein interaction (PPI) networks.** Genome-wide protein-protein interaction (PPI) datasets of *Saccharomyces cerevisiae* were taken from two sources:

(1) The high confidence PPI network of soluble yeast proteins derived by Collins et al.[55], built by merging interaction datasets (obtained using affinity purification/mass-spectrometry approaches) from two high-throughput studies by Krogan et al.[66] and Gavin et al.[67] The Collins et al. study used a Purification Enrichment (PE) statistical scoring system to identify high confidence interactions[55], resulting in 9070 interactions among 1622 distinct proteins. We also used the Collins dataset in a recent publication[68] and concluded that the network was of high quality.

(2) The multi-validated (MV), *S. cerevisiae* PPI network (version March 2018) from the BioGRID database[56]. These included literature-annotated PPIs detected using both high and low-throughput methods and included co-complex and binary interactions. Interaction reliability criteria were based on the number of different experimental techniques used to detect each physical interaction and the number of publications reporting the interaction. This dataset contained 14,285 interactions among 3737 proteins, and similar to the Collins network was curated for our recent publication[68].

**Generation of randomized PPI networks.** Ten different, random PPI networks were generated based on each of the Collins and BioGRID PPI networks, preserving the total number of protein nodes and the node degree distribution of each input network. The same protocol was also used in our recent publication[68]. First, binary interactions (i.e., edges) in the input network were shuffled randomly using the *sample_degseq* function of the *igraph* R package (http://www.igraph.org/r), avoiding repeated interactions and self-loops. Next, the node identities (i.e., protein ids) were randomly shuffled using the *sample* function, performed without replacement, using the R base package (http://www.r-project.org). Finally, the resulting networks were edited to eliminate any (few) remaining interactions that were also present in the original (starting) PPI network.

**Half-life datasets.** Two recent half-life datasets from *S. cerevisiae* were used from: (1) Martin-Perez and Villen[69] and, (2) Christiano et al.[70] From the Martin-Perez dataset, we obtained relative protein half-life values (half-life normalized by cell growth) averaged over two replicates for the prototrophic strain. From the Christiano dataset, protein half-lives (provided in minutes) were obtained.

**Gene Ontology (GO)-based semantic similarity calculation.** Functional similarity between two interacting proteins was evaluated using the GO semantic similarity measure. Functional annotations were obtained from the Gene Ontology[71] (July 2016). Only GO terms based on the evidence codes 'IDA' (Inferred from Direct Assay) and 'IPI' (Inferred from Physical Interaction) from the Biological Process (BP) ontology were considered. We used the semantic similarity measure developed by Wang et al.[72], available as an implementation in the R-package, GOSemSim[73]. The GO BP SemSim score provides a numeric estimate of the similarity of two sets of GO terms (corresponding to the functional annotation of two proteins) based on their location in the GO hierarchical graph and relationships to ancestor terms.

**Gene co-expression data.** Pearson's correlation coefficients calculated between mRNA expression profiles of *S. cerevisiae* gene pairs, obtained from COXPRESdb[74], were used to represent gene co-expression levels (1.0 indicates highly correlated co-expression). COXPRESdb computes mRNA expression profiles from expression (microarray) data measured under different conditions.

**PPI network paths.** The Collins network was input into the NetworkX (Network Analysis) Python package (https://networkx.github.io) as a graph object. Then the *all_pairs_shortest_path* algorithm was run on the graph to define shortest paths between all pairs of nodes (proteins). The output paths provide information on the starting and final members as well as intermediate path members (proteins).

**Statistical tests.** All statistical tests for calculating statistical significance (*P*-values) between groups were performed using the Mann-Whitney U-test (two-sided), unless otherwise specified.

**Reporting summary.** Further information on research design is available in the Nature Research Reporting Summary linked to this article.

## Data availability
All the relevant data(sets) are included in the manuscript and the supplementary information files.

## Code availability
Code (Python and R scripts) used to analyze the datasets and to generate the figures in this paper are available from the corresponding authors upon request.

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

## Acknowledgements

This work was supported by the Odysseus grant G.0029.12 from Research Foundation Flanders (FWO), a VUB Spearhead grant (SRP51, 2019–24), European Union's Horizon 2020 research and innovation programme grants under the Marie Skłodowska-Curie scheme (IDPfun, No. 778247) and WIDESPREAD-2020-5 Twinning scheme (PhasAge, No. 952334), grants K124670, K131702 from the Hungarian Scientific Research Fund (OTKA), a Mexican National Council of Science and Technology (CONACYT) PhD Fellowship [215503] (to MM-C) and a VIB/Marie Curie COFUND Postdoctoral (omics@VIB) fellowship (to MG).

## Author contributions

M.G. initially conceived the project. M.G. and P.T. designed the analyses and guided the project. M.G., T.L. and M.M.-C. performed the data curation and data analyses. All the authors analyzed the results. M.G. and P.T. wrote the manuscript with feedback from T.L. and M.M.-C. All the authors read and approved the manuscript.

## Competing interests

The authors declare no competing interests.
