## [Peer Review File · Communications Biology]

Reviewers' comments:

Reviewer #1 (Remarks to the Author):

Previously this group has published the tripartite degron principle, claiming that there is a primary degron, a motif recognized by E3 ubiquitin ligases, secondary site(s) comprising ubiquitinated lysine(s) (Ubsite) and a juxta positioned tertiary element of disordered segment. Here, based on systematic analysis (about 200 substrates), they propose that these degron elements are blocked by degron masking mechanism mediated by protein-protein interaction (PPI). Based on yeast data sets analysis, they propose the term degronons for co-regulation, at the protein stability level, of functional network modules. The degron masking principle was previously experimentally demonstrated in a number of studies however a systematic analysis of this regulatory mechanism has certain novel aspects of interest to others.

Comments

1. Systematic analysis is important in improving our predictions in identifying new cases lacking experimental evidence. The manuscript would be profoundly improved by highlighting and providing experimental validating for at list one of the predicted example.
2. The tripartite story is a bit confusing. Clearly the E3 binding motif is the dominant player and its masking is logical but the rationale behind targeting others, namely Ubsite or the tertiary element, is not clear enough (double locking only?).
3. The function of the masking partners was not annotated to show whether they exhibit a kind of tendency towards similar function with the interacting substrates.
4. The degronons model is based on the study of PPI and proteins half-lives, but not on degron masking principle. As the authors are aware of other mechanisms of regulation of protein degradation and do not think that masking degrons is the only mechanism, I found the degronons model too speculative at this stage.

Reviewer #2 (Remarks to the Author):

This manuscript describes results of systematic, meticulous, and well-designed study of the phenomenon of degron masking, which is a major mechanism utilized by the cellular systems to coordinate and regulate degron recognition and protein degradation. This work adds significantly to the field and will have a noticeable impact. The study is original and provides long needed answer to an important question on the potential regulation mechanisms of the degradation of proteins with intrinsically disordered regions (IDRs). This work provides strong evidence in support of the authors' hypothesis that "the structural plasticity of IDRs that enable embedding a dense network of overlapping interaction sites, can regulate protein turnover and exert dynamic control over the coordination of cellular pathways."

Congratulations for conducting this interesting and important study.

Vladimir N. Uversky

Reviewer #3 (Remarks to the Author):

Review Guharoy et al. Communications Biology

Running Title: Degron masking within protein interfaces

Title: Degron masking outlines degronons, co-degrading functional modules in the proteome

The authors carried out a systematic analysis and combined datasets from experimentally

validated degrons, onto which they mapped experimentally derived partner binding sites to explore potential degron masking. Guharoy et al proposed functionally and physically interconnected P-P interaction modules, degronons, have mutually correlated turnover (degradation) rates.

Comments:

1. In abstract, existence of (co-)degradation modules, abrupt as is, refrain from using it in the abstract?
2. Of note, Figure 1A-B is exactly the same schematic view already published in Guharoy et al. 2016 Nature Communications „Tripartite degrons confer diversity and specificity on regulated protein degradation in the ubiquitin-proteasome system. Can the authors devise updating the Figure particularly the separation of the three degron parts, since the authors mention a close connection between IDRs and Ub sites and degrons that are partially at least located in the IDR, which might likely serve as the initiation site at the proteasome (e.g. line 192). Figure c-e is an extension of the Figure, also part of an update?
3. Figure 5. A) is a bar plot in green and red. A no go for color/blind people. C) a heat map also in green and red. Please change colors.
4. Lines 44-57 too wordy, information without a context.
5. Figure 1E, please specify ELM (eukaryotic linear motifs) in text.
6. Issue with Fig 3a. Mixing in 1 graph info on 1 degron, Ub sites and IDRs vs. their overlap with ELMs counts might be misleading, since comparison of specific sites with motifs/domains
7. Issue with Fig 4B and statement. „Overall, E3s were significantly lower in abundance compared to substrates ($P=1.25E-12$), and, more strikingly, E3s were ~ 10 -fold less abundant than APs ($P=1.13E-62$). And cellular concentrations of E3s are limiting relative to APs; therefore, APs should exert strong regulatory effects by masking degrons for a substantial fraction of substrate lifetime.“ Is the statistical significance real?
8. Are ranks in Figure 5c inverted? If P53 is very low abundant in wild type the signal should be green for less abundant. But P53 accumulates in all cancer tissues but in HEK293, so it is high abundant, therefore it should be red with a score close to 100, while in HEK293 cells it should be low abundant in green with a score close to 100. In Figure 5c, either the colors in the heatmap, or the colors of the guide at the right seem to be inverted.
9. Be cautious with the following statement in line 396 „functional interacting modules are synthesized and degraded together. These observations were independent of the PPI network, as both the Collins and BioGRID networks showed similar profiles.
10. For clarity, please spell out „abundance ratios“ prior statement in lines 398-400 that reads „In contrast, abundance ratios of PPI pairs showed a less dramatic increase as a function of the T1/2 ratio category (Fig. 6e), suggesting that subunit stoichiometry may be largely independent of component half-lives.
11. The authors oversimplify the contribution of PP interaction kinetics. For instance, when they state „such assembly principles regulate subunit stoichiometries since ‘excess’ proteins, not incorporated into functional complexes, are degraded to avoid potential negative effects. This behavior points to likely degron unmasking upon subunit dissociation (and for unassembled subunits).“ The events are not binary and mutually exclusive. Slight changes in binding affinities can enhance or diminish other PPIs, in a „dance of proteins“ of sorts.
12. The authors mention that degron shielding by interaction partners different than the ones from the UPS components may increase proteins half-life, for this claim I have the following comments: Since proteins engage and disengage in dynamic and diverse P-P interactions often determined by affinities that are not exclusive and/or restricted to short degron motif, my main concern about the concept of degron shielding is that the authors did not use experimentally determined binding affinities in their data set (Table 1) neither with the E3-ligase nor with degronon components. Thus, while degron motifs may have multiple interaction partners, region outside the degron may or may not be promoting the binding to a specific partner from the UPS. Without adding a comprehensive analysis of binding affinities of the degradation substrates with E3s or alternative partners (APs) the degron shielding concept is a little bit weak. Along the same lines, this reviewer has an issue with the following statement: „To assess binding competition and estimate relative motif occupancies, we analyzed cellular abundances of the protein groups (substrates, E3s, and APs). Competition should be predominantly driven by relative abundances since the motif lengths are comparable and motif-based interactions are mostly weak and transient (K_d typically in the low μM range).“

The concept of protein binding kinetics is also disregarded in the section "Protein abundances and binding competition in degradation-regulatory modules" (page 12), although the authors determined that the abundance of E3_{substrate}APs (Figure 4b, 4c and 4d), they assume that competition for binding partners only comes from protein abundance since the affinity of a protein degron motif is the same for E3s and APs. This may not be true as regions outside the degron could increase the affinity of a protein either for E3s or for alternative partners. Additionally, their results in Figure 4c, show that their highly abundant E3s as much as APs, in this scenario when both E3s and APs are equally abundant the binding affinities, will determine which interaction will be favored. For this section, I would ask the author to relate the protein abundance with experimentally determined binding affinities available in the literature since their data set includes very well-characterized proteins.

13. The authors refer to „collective stability“, what about „collective destabilization“ considering the potential of a specific E3 targeting multiple substrates in the same regulatory network? The authors certainly adopt a „substrate“ perspective, but what about if one considers the E3 side? And the time window in which a degron is encompassed? Keeping in mind substrate residency on E3, an binding kinetics, as well as substrate turnover rate, it is possible an active E3 is not limited by its low levels, and does not require undergoing autoubiquitylation to move effectively to the next substrate. How do you incorporate into your model this seemingly „untuned“ scenario?

14. This reviewer agrees with the authors that the fact that degrons remain largely uncharacterized in eukaryotic proteins, and the lack of structural information of IDRs among others, one can only hypothesize the scenarios the authors mention. Therefore, although I find the various concepts the authors present of tremendous intellectual value, the lack of substantial experimental data available for the authors to analyze (authors carried out systematic analysis of publicly available data) makes this manuscript on degron masking rather conjectural and theoretical. If the authors adhere to this premise, and carefully state their concepts, the manuscript will be inspiring for future studies.

Other specific comments:

15. When first time p53 is mentioned (line 141) please mention it's synonymous for TP53, which is quite outstanding in Figure, 2 and should therefore be connected with the text. It would be beneficial if only one term is used throughout the manuscript

16. Given the number of up to hundreds of interaction partners for a single protein in Figure 2a-c, how well are those interactions studied and verified to be specific direct interactions and no indirect as e.g. by co-purification during a pulldown experiment?

17. In line 133 it is written that UPS-related partners were filtered out based on GO terms. However, later MDM2 (clearly a component of the UPS) will be discussed as a target being influenced by USP7 docking, which is itself part of the UPS from my point of view. Please comment. Is GO term annotation really a good filter?

18. Lines 160-163: The masking of the lysine sound quite speculative. Isn't there a better example? Or is the amino group of that lysine itself engaged in surface interaction to strengthen the point?

19. The masking of IDRs would require that those IDRs remain bound constantly and are not flexibly moving, is this right?

20. Line 241-247: Is USP docking really masking the tertiary degron, which is with ~60 amino acids very long and would possibly still be accessible to the proteasome, which needs only >23 amino acids as terminal initiation sites? The MDY2 example from Figure 2 is more plausible from my perspective. Please comment.

21. Line 277-279: Quite general statement, which can be easily challenged by multiple high-affinity degrons e.g. during auxin signaling and a comparable length is likely not given for IDRs as initiation sites, which are likely not meant here as motifs.

22. Figure 4 would benefit from an example showing the coupling of direct interacting AP, substrate and E3 ligase.

23. Figure 5: How can p53 be very abundant in the whole organism data set, but is not detectable in any tissue, besides at low level in heart tissue? An explanation should be added to the text lines 332-349; Figure 5b (left panel) is very hard to read, and is there perhaps a better depiction that helps to make the point.

This is one of the weakest sections of the manuscript relating to the tissue-specific and disease-

linked variation of PPI modules based on component abundances.

24. Lines 363-371: Was this dataset also used for an analysis like depicted in Figure 2 to have higher confidence on direct interactions, which would be the only one that could mask the degrons?

25. Discussion: Overall nice and a good incorporation of e.g. non-Ub mediated degradation at the 20S proteasome. However, a critical assessment of the presented analysis and especially the data used is missing in my opinion, e.g. why was a cutoff of 0.3 used for the analysis of database derived PPIs interactions and what is its influence? 0.3 seems to be very low and a more stringent analysis could benefit to identify more relevant true interactions even though the networks would be less big and impactful.

26. Methods section contain discussion statements, could they be integrated in the discussion section? Any particular reason for why the authors opted for keeping these in the Methods?

27. Methods; Lines 596-598: Is this really necessary? Would a small proportion of present interaction make a big difference or is this done simply to be extra safe? Might it increase bias?

Reviewers' comments and point-by-point author responses

Reviewer #1 (Expertise: Protein degradation and Intrinsically disordered proteins)

Previously this group has published the tripartite degron principle, claiming that there is a primary degron, a motif recognized by E3 ubiquitin ligases, secondary site(s) comprising ubiquitinated lysine(s) (Ubsite) and a juxta positioned tertiary element of disordered segment. Here, based on systematic analysis (about 200 substrates), they propose that these degron elements are blocked by degron masking mechanism mediated by protein-protein interaction (PPI). Based on yeast data sets analysis, they propose the term degronons for co-regulation, at the protein stability level, of functional network modules. The degron masking principle was previously experimentally demonstrated in a number of studies however a systematic analysis of this regulatory mechanism has certain novel aspects of interest to others.

We thank the reviewer for their comments and for appreciating and highlighting the fact that a systematic analysis of degron masking will be of interest and relevance to other researchers.

Comments

1. Systematic analysis is important in improving our predictions in identifying new cases lacking experimental evidence. The manuscript would be profoundly improved by highlighting and providing experimental validation for at least one of the predicted example.

We thank the reviewer for the comment. However, we, the co-authors on this manuscript, are all computational biologists and although a few (other) members of the Tompa group perform wet-lab experiments, none of them are experienced in experimental setups involving the UPS; nor are such setups readily available within the group or in the department. Therefore, although we appreciate the suggestion to perform experimental validation of one (or a few) cases, these experiments would be technically very difficult for us.

Indeed, that is precisely why we started this analysis by performing a large-scale literature review of experimentally verified cases where it was demonstrated that specific binding partners stabilize substrate half-life, even in the presence of the cognate E3 ligase (cf. data in Supplementary Table 1). For a subset of these cases, the exact location of the degron(s) were also identified and demonstrated to overlap with the partner binding sites. Given all these literature-derived examples, we feel that the experimental validation of one or two additional cases (as part of the current manuscript) would not broaden the experimental foundation and/or substantially change the outcome of the degron masking model that we have described. However, we have employed a broad range of available experimental datasets to develop and validate a PPI-based degron masking hypothesis that we expect will help answer open questions on the regulation of substrate targeting in the UPS. Further, we are convinced that this will inspire a range of experimental studies and guide wet-lab colleagues in discovering novel, exciting cases. We have included a sentence to this effect in the first paragraph of the Discussion. Please also note that we have agreed with the editor that we do not need to incorporate new experimental data as part of this revision.

2. The tripartite story is a bit confusing. Clearly the E3 binding motif is the dominant player and its masking is logical but the rationale behind targeting others, namely Ubsite or the tertiary element, is not clear enough (double locking only?).

The details of the tripartite degron model were presented in our earlier paper, Guharoy et al., Nat Comm 2016; 7: 10239. Here, we briefly reintroduced the key elements of the model in the Introduction (paragraph 2) and included a schematic outline of the elements in Figures 1a,b. The primary degron, i.e., the E3 binding motif is indeed an important player, conferring a major part of the specificity of substrate recruitment. However, Ubsites and degradation initiation segments are also clearly important for a complete, functional degradation motif (degron) and for a successful completion of the degradation cycle. Therefore, we have included examples of masking of all (either singly or in combination) of the tripartite degron elements that may independently regulate degradation and therefore not via simply additional locking. We have added a new paragraph in the Discussion (page 22) to clarify that the cooperative and successive action of all three degron elements is required for specific substrate targeting, and that the independent regulatory masking of each of these elements is possible (please also see reply to comment #3 below).

3. The function of the masking partners was not annotated to show whether they exhibit a kind of tendency towards similar function with the interacting substrates.

We thank the reviewer for this point. Taking the proteins analyzed in Figure 2, we have now compared the functional relationship between substrates and their degron masking partners. We have added a new supplementary Figure (Supplementary Figure 3) where we quantified (based on GO Biological Process term comparisons, described in Methods) the pairwise functional similarity between substrate-all partner pairs (numbers shown along the x-axis, Figure 2) and substrate-degron masking partner pairs (y-axis, Figure 2). We expected both groups of protein pairs to exhibit high functional similarity scores, since these are all high confidence, experimentally verified, physically interacting proteins. Hence, they must share similar cellular functions and participate in common pathways. Interestingly, the latter group (i.e., the degron masking partners) showed significantly higher functional similarity to the corresponding substrates (whose degrons they mask) as compared to general interacting partners of those substrates (new Suppl figure 3). Therefore, this strongly indicates that protein stability is extremely tightly regulated and that degron masking partners must constitute, in a sense, a 'special' subset of a substrate protein's interactome. Furthermore, this behavior was found to be true for the masking partners of *all* three degron types, strongly suggestive of the importance of masking all degrons (primary, secondary, and tertiary). Reflecting also on the previous comment #2, this GO analysis provides additional evidence that the masking of secondary and tertiary degrons may not be simply additional locking. Masking of these elements should be equally important as masking the primary degron. A new paragraph outlining the results of this analysis has been added to the results section (page 8).

4. The degronons model is based on the study of PPI and proteins half-lives, but not on degron masking principle. As the authors are aware of other mechanisms of regulation of protein degradation and do

not think that masking degrons is the only mechanism, I found the degretonons model too speculative at this stage.

This is true that the degretonon model is based on the analysis of yeast PPI datasets (Figure 6) in terms of protein half-life values and functional similarity analyses. Since degrons remain unknown/unidentified on a proteome-wide scale, we used the strong correlation observed between direct PPI partners and their strong half-life similarity as a proxy to hypothesize a potential co-stabilization mechanism. This is clearly mentioned in the final paragraph of Discussion.

Reviewer #2 (Expertise: Intrinsically disordered proteins, protein structure)

This manuscript describes results of systematic, meticulous, and well-designed study of the phenomenon of degreton masking, which is a major mechanism utilized by the cellular systems to coordinate and regulate degreton recognition and protein degradation. This work adds significantly to the field and will have a noticeable impact. The study is original and provides long needed answer to an important question on the potential regulation mechanisms of the degradation of proteins with intrinsically disordered regions (IDRs). This work provides strong evidence in support of the authors' hypothesis that "the structural plasticity of IDRs that enable embedding a dense network of overlapping interaction sites, can regulate protein turnover and exert dynamic control over the coordination of cellular pathways."

Congratulations for conducting this interesting and important study.

Vladimir N. Uversky

We thank Dr. Uversky for his highly appreciative and encouraging comments on our manuscript.

Reviewer #3 (Expertise: Intrinsically disordered proteins, degrons & protein stability)

Review Guharoy et al. Communications Biology

Running Title: Degreton masking within protein interfaces

Title: Degreton masking outlines degretonons, co-degrading functional modules in the proteome

The authors carried out a systematic analysis and combined datasets from experimentally validated degrons, onto which they mapped experimentally derived partner binding sites to explore potential degreton masking. Guharoy et al proposed functionally and physically interconnected P-P interaction modules, degretonons, have mutually correlated turnover (degradation) rates.

We thank the reviewer for their extensive and detailed comments, which have helped us to considerably improve the manuscript.

Comments:

1. In abstract, existence of (co-)degradation modules, abrupt as is, refrain from using it in the abstract?

We have appropriately re-phrased the second sentence of the abstract.

2. Of note, Figure 1A-B is exactly the same schematic view already published in Guharoy et al. 2016 Nature Communications „Tripartite degrons confer diversity and specificity on regulated protein degradation in the ubiquitin-proteasome system. Can the authors devise updating the Figure particularly the separation of the three degron parts, since the authors mention a close connection between IDRs and Ub sites and degrons that are partially at least located in the IDR, which might likely serve as the initiation site at the proteasome (e.g. line 192). Figure c-e is an extension of the Figure, also part of an update?

The original schematic figure from Guharoy et al., 2016 Nature Communications, depicting the essence of the tripartite nature of the degron, is central to our current analysis. Therefore, we have re-used it here (Figs. 1a and 1b), and this is clearly indicated in the figure legend. Figs. 1c-e, on the other hand, are used to explain the degron masking concept and schematically outline the methodological approach we have employed to analyze the phenomenon in this current manuscript.

In terms of an update to panels 1a and b, we still feel that this representation best captures the features of the tripartite degron constituents, as analyzed in Guharoy et al. Nat. Comm. 2016. For instance, primary degrons (yellow) were almost entirely located within IDRs, as depicted. Degradation-linked Ubsites were shared between IDRs and surface exposed regions of structured regions, but almost always proximal to IDRs (tertiary degron serving for proteasomal entry and degradation initiation). For that reason, we placed the Ubsites (orange) in the schematic right next to the tertiary degron (IDR shown in red). Therefore, we feel that this representation still captures all these characteristics in an optimum manner.

3. Figure 5. A) is a bar plot in green and red. A no go for color/blind people. C) a heat map also in green and red. Please change colors.

Thank you. We have modified the colors to dark grey and light grey for Fig. 5a (and the related Supplementary Figure 11). For the heatmap (Fig. 5c) we have modified the color palette to Red-Yellow-Blue.

4. Lines 44-57 too wordy, information without a context.

We have edited and simplified the first paragraph of Introduction, to better convey the message of the concept of coordinated protein degradation in the cell, and possible ways of studying its mechanism(s).

5. Figure 1E, please specify ELM (eukaryotic linear motifs) in text.

ELM has been defined upon first use in the main text, on page 10, first paragraph. We have also defined it in the Fig. 1e legend.

6. Issue with Fig 3a. Mixing in 1 graph info on 1 degron, Ub sites and IDRs vs. their overlap with ELMs counts might be misleading, since comparison of specific sites with motifs/domains

We put the three degnon types together in one plot (Fig. 3a) since we were interested to analyze degnon-ELM overlaps, considering all degnon types (irrespective of their nature: peptide motif, Ubsite, or IDR) as a single 'unit'. It is true that simply because of their different lengths the chances for a tertiary degnon to overlap with ELMs is expected to be higher than for a secondary degnon. However, this is not the case here, and moreover, we are *not* making any numerical/statistical comparisons such as whether, for example, docking sites tend to mask the secondary degnons more than the tertiary degnons. Finally, as we have clearly mentioned in the text, since experimentally validated ELMs are still limited in number, this is not really a statistical analysis, but more about (current) information showing the existence of ELMs overlapping with degnons. This is now indicated in a minor addition, last two sentences of the first paragraph, page 10.

7. Issue with Fig 4B and statement. „Overall, E3s were significantly lower in abundance compared to substrates ($P=1.25E-12$), and, more strikingly, E3s were ~10-fold less abundant than APs ($P=1.13E-62$). And cellular concentrations of E3s are limiting relative to APs; therefore, APs should exert strong regulatory effects by masking degnons for a substantial fraction of substrate lifetime.“ Is the statistical significance real?

Figure 4b is plotted on a log scale (y-axis), and as indicated by the y-axis labels, each horizontal line is a 10-fold value increase. Hence the differences as mentioned in the text are indeed statistically significant.

8. Are ranks in Figure 5c inverted? If P53 is very low abundant in wild type the signal should be green for less abundant. But P53 accumulates in all cancer tissues but in HEK293, so it is high abundant, therefore it should be red with a score close to 100, while in HEK293 cells it should be low abundance in green with a score close to 100. In Figure 5c, either the colors in the heatmap, or the colors of the guide at the right seem to be inverted.

The ranks and heatmap colors in Figure 5c are correct.

Firstly, to clarify better the data presented in the plot: we converted protein abundances (obtained as ppm values from PaxDB) into ranked abundances (rank bins). The rank bin conversion was done separately for each dataset, providing a numerical value for the position of each protein within each specific abundance dataset. In other words, is a protein within the top 1% most abundant (rank 1), top 5% (rank 5), 10% (rank 10), 20% (rank 20), and so on, amongst all the proteins measured in that dataset. In Figure 5c, each column corresponds to a specific abundance dataset. Higher ppm values will translate into low rank bin values (all details are also provided in the Methods section). Accordingly, a darker red color on the heatmap indicates higher protein ppm abundance (rank closer to 1) and a darker blue shade indicates lower abundance (rank closer to 100), as shown by the color scale. Please also note that we have changed the colormap in the new version of this figure, as per comment #3 of this reviewer. Red-Yellow-Green color palette used in the previous version now becomes Red-Yellow-Blue.

On the specific question of P53 (top row, Fig. 5c), P53 has an abundance of 17.6 ppm in the *H.sapiens* - Whole organism (Integrated) dataset (first column of the heatmap) and this converted to an abundance rank of 19 (plotted as dark orange). Whereas, in all the tissue datasets (except HEART integrated dataset), no P53 abundance measurements were available; hence, the respective cells are colored white

(i.e., no data). However, when cancer cell line datasets were considered, P53 becomes measurable whereas in the normal tissue datasets they were absent. For example, P53 abundances in U2OS, GAMG and HeLa cells were 36.3 ppm (rank 72), 45.1 ppm (rank 66) and 35.2 ppm (rank 70), respectively. Similarly, for the rest of the cancer cell lines, P53 was measurable but it ranked near the bottom (rank 90-100) within each of those datasets, hence the dark blue color. In other words, P53 goes from being unmeasurable (tissues datasets) to measurable (but low value) abundances in cancer cell lines.

Concerning the HEK293 cell line (last column on the right), although non-cancerous (but with a complex karyotype), we nevertheless kept it in the figure, since it is from the same publication (Geiger et al., MCP, 2012; [https://www.mcponline.org/article/S1535-9476\(20\)30500-4/fulltext](https://www.mcponline.org/article/S1535-9476(20)30500-4/fulltext)) as the other 10 cancer cell lines (indicated at the bottom of the figure). In this HEK293 dataset, the abundance of P53 was measured to be 164 ppm (much higher than the measured P53 abundances in the other cancer cells), and therefore, relative to the other proteins in the HEK293 dataset, it was ranked 16.

We have made further edits to the appropriate Results and Methods sections as well as the legend of Figure 5c to better clarify these points.

9. Be cautious with the following statement in line 396 „functional interacting modules are synthesized and degraded together. These observations were independent of the PPI network, as both the Collins and BioGRID networks showed similar profiles.”

Thank you, this statement was too direct and pointed a little beyond what we have presented evidence for: we have removed the critical first sentence.

10. For clarity, please spell out „abundance ratios“ prior statement in lines 398-400 that reads „In contrast, abundance ratios of PPI pairs showed a less dramatic increase as a function of the T1/2 ratio category (Fig. 6e), suggesting that subunit stoichiometry may be largely independent of component half-lives.”

This is now more clearly worded to define what we mean by abundance ratios.

11. The authors oversimplify the contribution of PP interaction kinetics. For instance, when they state „such assembly principles regulate subunit stoichiometries since ‘excess’ proteins, not incorporated into functional complexes, are degraded to avoid potential negative effects. This behavior points to likely degon unmasking upon subunit dissociation (and for unassembled subunits).“ The events are not binary and mutually exclusive. Slight changes in binding affinities can enhance or diminish other PPIs, in a „dance of proteins“ of sorts.

We completely agree that beyond K_d and protein concentration (abundances), PP interaction kinetics will have a basic influence on the regulatory outcome. We attempted to take it into consideration, but such data are extremely scarce, and are (being very sensitive to experimental conditions) probably much less reliable than thermodynamic values. To clarify this point, we have added a few explanatory sentences to the relevant section of Discussion (page 23).

12/1. The authors mention that degron shielding by interaction partners different than the ones from the UPS components may increase proteins half-life, for this claim I have the following comments: Since proteins engage and disengage in dynamic and diverse P-P interactions often determined by affinities that are not exclusive and/or restricted to short degron motif, my main concern about the concept of degron shielding is that the authors did not use experimentally determined binding affinities in their data set (Table 1) neither with the E3-ligase nor with degronon components. Thus, while degron motifs may have multiple interaction partners, regions outside the degron may or may not be promoting the binding to a specific partner from the UPS. Without adding a comprehensive analysis of binding affinities of the degradation substrates with E3s or alternative partners (APs) the degron shielding concept is a little bit weak.

We agree with this point. For several protein systems (including those presented in Table 1), we have searched for and collected Kd data, wherever available, to compare substrate-E3 ligase and substrate-alternative partner (AP) binding interactions (new Supplementary Table 10 and Supplementary Figure 7). Comparison of the Kd value distributions (Supplementary Figure 7a) shows that the dissociation constants for substrate-AP interactions are overall similar to those for substrate-E3s. On the other hand, abundances of APs are often greater than E3s (Figure 4b and Supplementary Figure 7b), which would make the binding of APs and therefore masking substrate degrons from E3 more probable. This additional analysis of Kds is now described in the section "Protein abundances and binding competition in degradation-regulatory modules" (paragraph 1, page 13 and page 14).

12/2. Along the same lines, this reviewer has an issue with the following statement: „To assess binding competition and estimate relative motif occupancies, we analyzed cellular abundances of the protein groups (substrates, E3s, and APs). Competition should be predominantly driven by relative abundances since the motif lengths are comparable and motif-based interactions are mostly weak and transient (Kd typically in the low uM range).“

We believe that our reply to the previous point covers this related point as well.

12/3. The concept of protein binding kinetics is also disregarded in the section "Protein abundances and binding competition in degradation-regulatory modules" (page 12), although the authors determined that the abundance of E3<substrate<APs (Figure 4b, 4c and 4d), they assume that competition for binding partners only comes from protein abundance since the affinity of a protein degron motif is the same for E3s and APs. This may not be true as regions outside the degron could increase the affinity of a protein either for E3s or for alternative partners.

Indeed, we have strengthened this analysis: firstly, we have addressed the relative Kds of substrate-E3 vs. substrate-AP binding, which strengthens our conclusion. Secondly, we fully agree that binding (and unbinding) kinetics of the different partners has an important effect in the regulatory outcome of protein-protein interactions, as already detailed under point #11. We have also stated this in the relevant paragraph of Discussion, on page 23.

12/4. Additionally, their results in Figure 4c, show that their highly abundant E3s as much as APs, in this scenario when both E3s and APs are equally abundant the binding affinities, will determine which interaction will be favored. For this section, I would ask the author to relate the protein abundance with

experimentally determined binding affinities available in the literature since their data set includes very well-characterized proteins.

Upon comparing the y-axes of Figures 4c and 4d, which are plotted on the same scale, we can already visually distinguish that the majority of APs (their abundances plotted on the y-axis, Figure 4d) are shifted to the upper range of abundances (>1 ppm). In contrast, the abundances of E3 ligases (y-axis, Figure 4c) are significantly downshifted towards lower values as compared to APs. This can also be clearly seen in Figure 4b, which compares the abundances of all the three groups (substrates, E3s and APs). Also keeping in view that these abundances are plotted on a log scale, these are significantly large differences in relative abundances of these protein groups (particularly the comparison between E3s and APs).

Taken in conjunction with the observation that overall, the distribution of substrate-E3 and substrate-AP dissociation constants (K_d) are not significantly different (new Supplementary Fig. 7a), this leads us to propose that cellular abundances are likely to play a major role. However, we fully agree that relating the protein abundances together with experimentally determined K_d s (wherever available) would be useful to analyze. We have presented this data in Supplementary Figure 7b. In this figure, for those systems where we obtained K_d data for both substrate-E3 and substrate-AP(s) binding, we see that the range of K_d values for AP interactions are either very similar to or smaller than the K_d s measured for E3 interactions (except for p53). Taken in conjunction with the abundances of the proteins (also shown on the plot), which shows that APs are more abundant than E3s, this gives greater confidence to the proposed model that APs will mask substrate degrons for a significant proportion of substrate lifetime. This is now stated on paragraph 1, page 14 of the Results.

13. The authors refer to „collective stability“, what about „collective destabilization“ considering the potential of a specific E3 targeting multiple substrates in the same regulatory network? The authors certainly adopt a „substrate“ perspective, but what about if one considers the E3 side? And the time window in which a degron is encompassed? Keeping in mind substrate residency on E3, and binding kinetics, as well as substrate turnover rate, it is possible an active E3 is not limited by its low levels, and does not require undergoing autoubiquitylation to move effectively to the next substrate. How do you incorporate into your model this seemingly „untuned“ scenario?

We certainly agree that E3s targeting multiple substrates can synchronize protein degradation. An E3-centric view is an interesting study that would merit a detailed future analysis. As outlined in this paper, however, the regulation of substrate stability via multiple degron-masking APs already creates potentially complex regulatory phenomena. Substrate-AP complexes should modulate substrate residency on E3 and via this mechanism disassociate E3 abundance as a limiting factor for substrate turnover, i.e., once degrons get unmasked, E3 activity would no longer be constrained by its typically low abundances. We have mentioned this on page 23 of the Discussion.

Concerning the “collective stability” of proteins (for example, subunits of a multiprotein complex or members of a conserved biological pathway), whereas the possibility of co-degraded functional units in the proteome (degrons) is appealing for its biological logic, we do appreciate the “collective destabilization” idea and have outlined the possibility of such a regulatory modality in a new section of Discussion, final paragraph on page 24.

14. This reviewer agrees with the authors that the fact that degrons remain largely uncharacterized in eukaryotic proteins, and the lack of structural information of IDRs among others, one can only hypothesize the scenarios the authors mention. Therefore, although I find the various concepts the authors present of tremendous intellectual value, the lack of substantial experimental data available for the authors to analyze (authors carried out systematic analysis of publicly available data) makes this manuscript on degron masking rather conjectural and theoretical. If the authors adhere to this premise, and carefully state their concepts, the manuscript will be inspiring for future studies.

We do appreciate that the reviewer assigns “tremendous intellectual value” to our findings and thinks that “the manuscript will be inspiring for future studies”. With reference to the “lack of experimental data”, we have appropriately re-written the relevant sections to emphasize that our insight into the degron space is rather limited (i.e., we don’t know the degrons of most substrates, and don’t know degrons targeted by most E3s). By analyzing different types of available experimental data, nonetheless, we have proposed a general regulatory model, which we anticipate inspiring future studies on substrates of interest and guide the design of dedicated experiments. We have made clear all these points in the revised Discussion.

Other specific comments:

15. When first time p53 is mentioned (line 141) please mention it’s synonymous for TP53, which is quite outstanding in Figure, 2 and should therefore be connected with the text. It would be beneficial if only one term is used throughout the manuscript

We have used p53 throughout the text and to avoid confusion added that TP53 is the gene name for human p53.

16. Given the number of up to hundreds of interaction partners for a single protein in Figure 2a-c, how well are those interactions studied and verified to be specific direct interactions and no indirect as e.g. by co-purification during a pulldown experiment?

As mentioned in the Methods section (“PPI binding site information”), the degron masking interactions (those along the y-axis in Figure 2) were identified from PPIs with detailed annotations, which indicate that these are high confidence, direct, physical interactions. Please see also the answer to comment #25 of this reviewer.

17. In line 133 it is written that UPS-related partners were filtered out based on GO terms. However, later MDM2 (clearly a component of the UPS) will be discussed as a target being influence by USP7 docking, which is itself part of the UPS from my point of view. Please comment. Is GO term annotation really a good filter?

This GO based filtering was applied to the IntAct-derived PPI data analysis presented in Figure 2. It works correctly, and if it was not applied then we would introduce noise. However, the USP7 discussion was in a different section of the paper where we analyzed degron-overlapping SLiMs and their binding partners. In this section (“Short interaction motifs overlapping with degrons ...”) we analyze manually curated alternative binding partners and not large-scale PPIs.

18. Lines 160-163: The masking of the lysine sound quite speculative. Isn't there a better example? Or is the amino group of that lysine itself engaged in surface interaction to strengthen the point?

We feel that it is a good example because it has X-ray crystallography data, so the proposition is not speculative. Lys is physically interacting with the partner protein, which should reduce its backbone flexibility.

19. The masking of IDRs would require that those IDRs remain bound constantly and are not flexibly moving, is this right?

This actually falls into a grey zone, as flexibility may allow modifications to happen even in the bound state (due to transient and partial dissociation). Also linked with the issue of binding-unbinding kinetics (question #11 of this reviewer). For the lack of appropriate data, this cannot be generally addressed and thus we accept that overlapping binding sites do result in shielding of the degron of IDP substrates.

20. Line 241-247: Is USP docking really masking the tertiary degron, which is with ~60 amino acids very long and would possibly still be accessible to the proteasome, which needs only >23 amino acids as terminal initiation sites? The MDY2 example from Figure 2 is more plausible from my perspective. Please comment.

Indeed, the MDY2 example from Figure 2f has been experimentally verified. Although in the case of Mdm2 (Figure 3d), the USP7 docking motif is only 5 residues, it can nevertheless be reasonably assumed that some motif flanking residues also interact with USP7, i.e., the amount of IDR accessible to the proteasome will be significantly reduced and hamper degradation initiation.

21. Line 277-279: Quite general statement, which can be easily challenged by multiple high-affinity degrons e.g. during auxin signaling and a comparable length is likely not given for IDRs as initiation sites, which are likely not meant here as motifs.

Indeed, in this section analyzing abundances of competing partners, we focus on primary degrons, which are typical SLiMs and their overlapping motifs, which have similar lengths. Now in the revised manuscript, we have added the analysis of dissociation constants (K_d data; Supplementary Table 10 and Supplementary Figure 7) and updated the corresponding Discussion on the comparison of K_ds of substrate interactions with E3 ligases and with degron-masking alternative partners (AP). Nevertheless, it is true that the presence of multiple, high affinity degrons might confer an exception; similarly, the relative binding strengths between tertiary degron-proteasome and APs will also be interesting to study, but such situations are beyond the scope of this paper.

22. Figure 4 would benefit from an example showing the coupling of direct interacting AP, substrate and E3 ligase.

In fact, the degradation regulatory systems analyzed in Figure 4 are all detailed in Table 1. All those systems were manually curated by us (using database and literature-based evidence) as forming direct coupling of substrate, E3 ligase and APs.

23. Figure 5: How can p53 be very abundant in the whole organism data set, but is not detectable in any tissue, besides at low level in heart tissue? An explanation should be added to the text lines 332-349;

Figure 5b (left panel) is very hard to read, and is there perhaps a better depiction that helps to make the point. This is one of the weakest sections of the manuscript relating to the tissue-specific and disease-linked variation of PPI modules based on component abundances.

In Figure 5c, firstly, these are not the full set of abundance datasets, only a subset. Of the total of 170 human protein abundance datasets obtained from PaxDb (see Methods and Supplementary Table 11), in figure 5c we plotted data from PaxDb “integrated” datasets (*left group*), PaxDb individual tissue datasets derived from Wilhelm et al. (*middle*) and PaxDb cancer cell line datasets derived from Geiger et al. (*right*). This is mentioned in the figure legend and also marked using the horizontal lines under Fig. 5c. PaxDb generates “integrated” datasets (left section, Fig. 5c) using a weighted averaging procedure at a whole-organism level (and for specific tissues) for which multiple experiments/datasets are available, leading to “best-estimate” datasets (<https://pubmed.ncbi.nlm.nih.gov/25656970/>). Therefore, the human Whole Organism integrated dataset (left most column, Fig. 5c) is based on other whole organism published datasets (see Supplementary Table 11) which are not present in the figure. It cannot be directly linked to the other tissue “integrated” datasets shown on the figure. Instead, the tissue-specific integrated datasets are in turn based on individual tissue experiments/datasets, which are not present in the figure. We wanted to visualize the full set of “integrated” datasets in this figure since, as stated above, PaxDb considers these as being the “best-estimate” datasets. We have added an explanatory sentence about PaxDb’s integrated datasets and the integration procedure on page 16 as well as in the Methods section, page 26.

About Figure 5b (*left panel*), the idea behind showing the dense P53 interactome was to highlight that using the methodology we proposed, by identifying specific interaction partners whose binding site(s) overlap and thereby shield degrons, we can select that particular PPI subset for correlative analysis of gene/protein expression (here using protein abundances). Of the large number of PPIs known for P53 (Fig. 5b, *left panel*), partners that bind overlapping the Mdm2-regulated primary degron of P53 (curated in Table 1 and highlighted on Fig. 5b) constitute the degradation regulatory network. The partners and their binding regions overlapping with the primary degron of P53 are then highlighted (*right panel*, Fig. 5b). This has now been clearly stated on page 17, second paragraph.

To conclude, the aim of this section of this manuscript is to outline (using an example) the concept of dissecting protein components within the interactome of a substrate protein, using binding site information and degron locations, and then analyzing abundance variations of the substrate, the E3 ligases(s) and the degradation regulatory components. Here, we selected a small subset of datasets to prepare this figure. As we have suggested clearly in the Discussion, this methodology can be extended in many directions and in myriad ways but here we outline and illustrate the concept and potential usages.

24. Lines 363-371: Was this dataset also used for an analysis like depicted in Figure 2 to have higher confidence on direct interactions, which would be the only one that could mask the degrons?

No, unlike the proteins in Figure 2 for which we have precise degron information, the analysis in Figure 6 uses yeast data (yeast-specific PPI networks, half-life measurements, protein abundances, gene expression data, etc.). Since we do not have the degron information for these proteins, we cannot perform an analysis analogous to Figure 2.

25. Discussion: Overall nice and a good incorporation of e.g. non-Ub mediated degradation at the 20S proteasome. However, a critical assessment of the presented analysis and especially the data used is missing in my opinion, e.g. why was a cutoff of 0.3 used for the analysis of database derived PPIs interactions and what is its influence? 0.3 seems to be very low and a more stringent analysis could benefit to identify more relevant true interactions even though the networks would be less big and impactful.

Thank you, it was indeed appropriate to mention non-Ub-mediated degradation in the Discussion since that pathway is also clearly regulated via PPIs. Regarding a critical assessment of the presented analysis, we have extensively edited the Discussion to include further assessment of our findings in the light of current knowledge (and potential future work).

Finally, the MI-score (Villaveces et al., Database (Oxford), 2015: bau131) cutoff of 0.3 used to filter out IntAct-derived PPIs was selected based on the following rationale. IntAct defines interaction confidence MI-scores based on all the available annotation evidence associated with an interaction. In general, the IntAct database regards data with a score of >0.6 as high-confidence and 0.45–0.6 as medium confidence. However, a significant fraction of true interactions will still have scores between 0.3-0.4 (an analysis is shown in Figure 6 of the Villaveces et al. paper). Also, please note that Supplementary Figure 6c in our paper plots the MI-score distribution for the full IntAct dataset, and not just those of the degron containing substrates. For the latter proteins, in order to identify the degron masking partners, we further queried for availability of detailed annotations (e.g., “sufficient binding region”, etc.). The full list of queried annotation terms is listed on pages 25 and 26. PPIs associated with such detailed annotations score more highly than less detailed ones. We have indicated this in the text, page 25.

26. Methods section contain discussion statements, could they be integrated in the discussion section? Any particular reason for why the authors opted for keeping these in the Methods?

In a few cases where the statements were focused on the technical side, we decided that it would be more appropriate to keep those in the Methods.

27. Methods; Lines 596-598: Is this really necessary? Would a small proportion of present interaction make a big difference or is this done simply to be extra safe? Might it increase bias?

As outlined in the Methods, the generated random PPI networks were checked for the presence of any real PPIs. This was indeed a very small proportion of the randomly generated interactions and would not influence the overall statistics. Nevertheless, we chose to remove these real interactions to ensure that the random networks were completely composed of non-native interactions. It should not increase bias at all, on the contrary, it should eliminate any minor remaining bias. The same procedure was used in another of our recent publications, Macossay-Castillo et al., *J Mol Biol.* 2019;431(8):1650-1670.

REVIEWERS' COMMENTS:

Reviewer #1 (Remarks to the Author):

Based on systematic analysis, the authors describe an exciting model of degron masking mechanism mediated by protein-protein interaction (PPI). The current revised version is much improved, and the original comments were effectively answered.